# Ketamine's acute effects on negative brain states are mediated through distinct altered states of consciousness in humans

Laura M. Hack[1,2,8], Xue Zhang[1,8], Boris D. Heifets [1,3], Trisha Suppes[1,4], Peter J. van Roessel [1,2], Jerome A. Yesavage[1,2], Nancy J. Gray[1], Rachel Hilton [1], Claire Bertrand[1], Carolyn I. Rodriguez [1,4,9], Karl Deisseroth [1,5,6,9], Brian Knutson [7,9] & Leanne M. Williams [1,2,9] ✉

Ketamine commonly and rapidly induces dissociative and other altered states of consciousness (ASCs) in humans. However, the neural mechanisms that contribute to these experiences remain unknown. We used functional neuroimaging to engage key regions of the brain's affective circuits during acute ketamine-induced ASCs within a randomized, multi-modal, placebo-controlled design examining placebo, 0.05 mg/kg ketamine, and 0.5 mg/kg ketamine in nonclinical adult participants (NCT03475277). Licensed clinicians monitored infusions for safety. Linear mixed effects models, analysis of variance, t-tests, and mediation models were used for statistical analyses. Our design enabled us to test our pre-specified primary and secondary endpoints, which were met: effects of ketamine across dose conditions on (1) emotional task-evoked brain activity, and (2) sub-components of dissociation and other ASCs. With this design, we also could disentangle which ketamine-induced affective brain states are dependent upon specific aspects of ASCs. Differently valenced ketamine-induced ASCs mediated opposing effects on right anterior insula activity. Participants experiencing relatively higher depersonalization induced by 0.5 mg/kg of ketamine showed relief from negative brain states (reduced task-evoked right anterior insula activity, 0.39 SD). In contrast, participants experiencing dissociative amnesia showed an exacerbation of insula activity (0.32 SD). These results in nonclinical participants may shed light on the mechanisms by which specific dissociative states predict response to ketamine in depressed individuals.

Racemic ketamine (1:1 esketamine: arketamine) is a non-competitive antagonist of the N-methyl-D-aspartate glutamate receptor and an FDA-approved dissociative anesthetic. It was introduced into clinical practice in the US in the 1960's as a novel agent that could produce effective sedation and analgesia while maintaining cardiorespiratory stability[1]. By the turn of the 21st century, research began to show that subanesthetic doses of ketamine had potential as a breakthrough, fast-

acting therapy for treatment-resistant depression (TRD)[2]. Throughout the 2000's, ketamine has increasingly been used as an off-label therapy for TRD, especially among suicidal patients[3]. Its use has increased since the US FDA approved intranasal esketamine for TRD in adults in March 2019 and for depressive symptoms in adults who have major depressive disorder with acute suicidal ideation or behavior in August 2020. Emerging evidence indicates that ketamine is also a promising therapy

for other psychiatric conditions, including substance use disorders (SUDs)[4], as well as post-traumatic stress, obsessive-compulsive, and social anxiety disorders[5].

The doses of ketamine used for these psychiatric disorders commonly induce acute dissociation and other altered states of consciousness (ASCs). While some of these non-ordinary states are associated with acute changes in effect as well as longer-term therapeutic affective changes, they can also cause unpleasant psychotomimetic experiences compared to placebo[6]. As a result, there have been efforts to design molecules with rapid-acting antidepressant properties, but without the psychedelic effects[7].

Mechanistic studies have demonstrated that ketamine acutely induces multiple changes in affective neural circuit activity[8]. However, it is not known whether affective neural changes are dependent on these non-ordinary subjective experiences. To address this knowledge gap, we used functional neuroimaging to engage key regions of the brain's affective circuits during acute ketamine-induced ASCs within a multi-modal, placebo-controlled design. This design enabled us to disentangle ketamine-induced effects on different subdomains of ASCs and to test which of these altered states mediate changes in neural circuit activity engaged by social affective stimuli.

Ketamine-induced dissociation can encompass a broad range of altered states that span the spectrum of aversive experiences, from relieving negative states to boosting positive feelings. Depersonalization and derealization involve disconnection from one's body and surroundings[9,10]. The ketamine-induced sensation of disconnecting from one's physical body may also be accompanied by detachment from negative affective and pain states, including emotional pain and depressive feelings[11]. In participants with depression, researchers have observed that a standard therapeutic ketamine infusion (0.5 mg/kg infused over 40 min) brings significant relief from overall depression[12], anhedonia[13], and suicidal ideation[14] during the same time frame within which dissociation occurs, which is as early as 5 min into the infusion with resolution typically within 1 h[5]. These observations of acute relief from overall depression[12] and anhedonia[13] can last from days to weeks.

Beyond relief from negative affective experiences, ketamine has also been observed to open the capacity for positive experiences, such as feelings of pleasure, love, and peace. On the other hand, ketamine has been reported to produce states experienced as aversive, such as hostility, fear, or horror related to paranoid thoughts, impairment in cognition, losing control, or gaps in memory (amnesia). Furthermore, dissociation is generally viewed as an adverse effect in the medical model of ketamine administration[16], yet debate remains regarding whether and how dissociative and other altered states are involved in ketamine's antidepressant effects[15]. Some studies have shown that particular aspects of dissociation and other ASCs are predictive of either response or nonresponse[17–19], which indicates the importance of separately assessing different components of these states to advance more targeted therapeutic ketamine strategies.

To unpack how ketamine-induced dissociative and altered states influence affective states, it is essential to understand the neural mechanisms that underlie these effects. In this study, we used blood-oxygen-level-dependent functional magnetic resonance imaging (BOLD fMRI) to assess neural activity engaged by positive and negative affective social stimuli following acute ketamine infusions that induced dissociative and altered states of consciousness. We focused on the anterior insula, amygdala, and anterior cingulate cortex (ACC), key regions in the brain's affective circuit established in randomized controlled designs[20, 21]. The amygdala responds automatically or implicitly to negative stimuli under the regulation of the subgenual ACC (sgACC)[22]. The anterior insula plays a role in subjective experiences that are the read-out of automatic bodily

states of emotional reaction[23], whereas the dorsal ACC (dACC) is involved in appraisal and expression of emotion[24]. The anterior insula, sgACC, and dACC also compose an emotional pain pathway that controls the emotional aspect of pain[11]. The nodes within this affective circuit are highly interconnected, and together they support the neural processes by which the human brain responds to negative affective stimuli and generates feelings about associated states. This affective circuit, including the anterior insula, amygdala, and ACC, is robustly activated in response to social stimuli such as facial expressions of emotion (for review, see ref. 25). The facial expressions of emotion task used in the present study utilizes social emotional stimuli, is modulated by intervention in disorders such as depression[21], and is well established for mechanistic target engagement trials[20].

The activity of amygdala, anterior insula, sgACC, and dACC in response to affective stimuli is the pertinent target for investigating the neural basis of ketamine-induced dissociative experiences such as depersonalization. There is emerging evidence that, relative to a non-placebo baseline condition, ketamine modulates anterior insula, amygdala, and ACC activity in response to facial emotion stimuli during BOLD fMRI[26–28]. Our focus on the affective circuit regions of interest also reflects their importance in psychiatric conditions now treated with ketamine, including depression and SUDs. Alterations in activity and connectivity of these negative circuit nodes are consistently observed in depression (for review, see ref. 24) and SUDs (for review, see ref. 29).

The present study design incorporated several key features. First, we assessed multiple aspects of ketamine-induced ASCs, integrated with BOLD fMRI, to probe positive and negative affective brain states using a tightly controlled experimental design that included double-blinding, randomization to limit carry-over effects, and a placebo control condition. Second, we infused ketamine or placebo over a period of 40 min, exactly matching the standard therapeutic designs. In contrast, prior mechanistic studies have tended to utilize a ketamine infusion protocol that involved infusing a bolus, then initializing a maintenance infusion to maintain blood concentrations at a stable level. We chose to use the standard therapeutic approach (0.5 mg/kg infused over 40 min) to facilitate the future clinical translation of the mechanistic findings. Third, due to our multi-modal approach, we were positioned to test whether specific aspects of ketamine-induced dissociation and other ASCs modulated acute changes in neural activity.

We had the following objectives: (1) to test whether specific aspects of dissociation and other ASCs differed across dose conditions; (2) to test the dose-dependent effects of ketamine on brain activity in response to emotional expressions; and (3) to test whether specific aspects of ketamine-induced dissociation and other ASCs mediated the effect of dose on acute changes in neural activity during emotional processing. We hypothesized that ketamine would induce multiple types of dissociative and altered experiences in a dose-dependent manner that mediate distinct effects on anterior insula and amygdala neural targets in response to affective stimuli. Specifically, we investigated whether experiences such as dissociative depersonalization, derealization, and altered states of bliss would mediate a reduction of anterior insula and amygdala activity, reflecting relief from negative affective states and the promotion of positive ones, whereas other states, including dissociative amnesia, anxiety, and impaired control and cognition, might instead increase anterior insula and amygdala activity, reflecting an exacerbation of negative affective brain states.

## Results

The current study compared outcomes between three drug conditions: placebo, 0.05 mg/kg ketamine, and 0.5 mg/kg ketamine. We hereafter refer to the results using only placebo and the 0.5 mg/kg

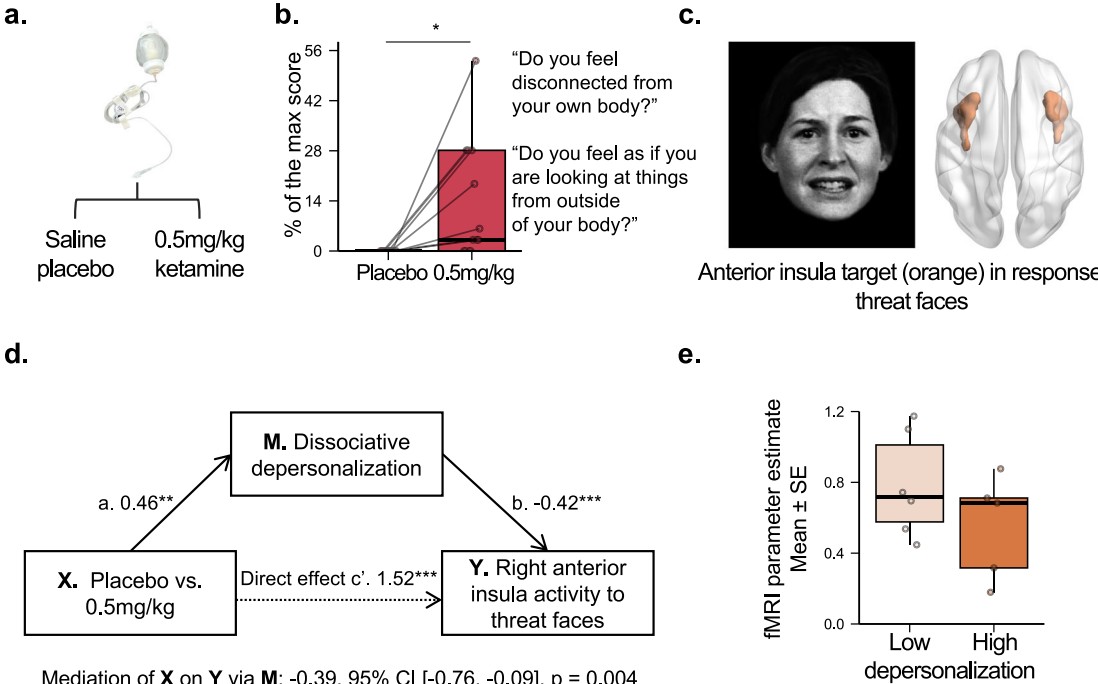

**Fig. 1 | Dissociative experiences of depersonalization mediate ketamine-induced decreases in insula activation in response to social threat.**
**a** Randomized dose (**X**) (*n* = 13 nonclinical participants). **b** Dissociative depersonalization (**M**) at end of infusion (40 min). Example items are listed. A two-sided post-hoc paired t-test was used to compare the effect of placebo vs. 0.5 mg/kg of ketamine on depersonalization ($T_{12}$ = 2.85, *p* = 0.02, Cohen's *d* = 0.82, 95% CI = [3.16, 23.30]). We implemented an FDR correction to control for the testing of multiple scale sub-components. **c** Anterior insula engaged by threat faces measured by fMRI (**Y**). The facial expression was extracted from Fig. 1 of Williams et al.[47]. The image was modified with permission from the developers of a database of 3D facial stimuli described in Gur et al.[73]. **d** Ketamine-induced effects (**X**) on reducing right anterior insula activity to threat faces (**Y**). An Averaged Causal Mediation Effect (ACME) mediation model was used to assess whether ketamine-induced dissociative depersonalization mediates the effect of dose on acute changes in neural activity during emotional processing. No adjustments for multiple comparisons were made. **e** Right anterior insula activity in response to threat faces as a function of depersonalization. In each boxplot, the central thick black line represents the median, color shaded boxes represent the first and third quartiles (the 25th and 75th percentiles), and the whiskers extend no further than 1.5 times of the distance between the first and third quartiles. *\**p* < 0.05, ***p* < 0.01, ****p* < 0.001. Source data are provided as a Source Data file. fMRI, functional magnetic resonance imaging. FDR, false discovery rate.

condition as "two conditions" (Figs. 1a and 2a), and the results using all three drug conditions as "three conditions" (Suppl. Fig. 1a).

**Ketamine increases dissociative depersonalization, dissociative derealization, and a blissful state in a dose-dependent manner**
As hypothesized, we observed a significant dose-dependent effect of ketamine in dissociative depersonalization ($F_{2,76}$ = 8.55, *p* < 0.001, the False Discovery Rate corrected p-value (pFDR) = 0.001; Fig. 1b for 0.5 mg/kg versus placebo, the two conditions, and Suppl. Figure 1d for 0.5 mg/kg, 0.05 mg/kg, and placebo, the three conditions) and dissociative derealization ($F_{2,76}$ = 13.97, *p* < 0.001, pFDR < 0.001; Suppl. Fig. 1e) as measured by the Clinician-Administered Dissociative States Scale (CADSS)[30]. Post-hoc paired *t*-tests between placebo and ketamine at 0.05 mg/kg and 0.5 mg/kg doses showed a significant increase only in the 0.5 mg/kg condition as compared to placebo for depersonalization ($T_{12}$ = 2.85, *p* = 0.02, Cohen's *d* = 0.82, 95% CI = [3.16, 23.30]; Suppl. Table 1), as well as derealization ($T_{12}$ = 3.69, *p* = 0.003, Cohen's *d* = 1.06, 95% CI = [5.75, 22.35]; Suppl. Table 1). Similarly, a significant dose-dependent effect was observed for a blissful state as measured by the 5-Dimensional Altered States of Consciousness (5D-ASC) rating scale[31,32] at the midpoint of the infusion (T = 20 mins, $F_{2,25}$ = 6.65, *p* = 0.004, pFDR = 0.02; Suppl. Fig. 1f), with the significant increase only in the 0.5 mg/kg condition as compared to placebo ($T_{12}$ = 2.62, *p* = 0.02, Cohen's *d* = 0.76, 95% CI = [3.45, 37.37]).

The experience of dissociation under the 0.5 mg/kg ketamine was also captured by free response reports from participants. We recorded narratives from five participants for all three drug conditions. Suppl.

Table 2 lists the quantitative ASC scores and free reports for each of the five subjects. During the infusion of 0.5 mg/kg ketamine−usually around 20−30 min into the infusion−three of these five participants endorsed altered spatial, visual, and auditory perceptions corresponding to the derealization dimension of dissociation. Relevant excerpts follow:

"I feel like I'm floating. I feel like I'm in a spaceship. Super disoriented−there is no north or south. I'm in the center and there's no bottom. Geometric shapes are felt, not seen." (P10)

"It feels very expansive, yet restrictive at the same time. Feels like I'm in a tunnel. Stretchy wobbly sensation. Space is stretching out and coming back." (P10)

"I'm feeling weird, my body is tingling, hearing a hum, especially in my right ear. I've never felt this way before, neutral color all around." (P11)

"Time felt stretched and that it would take time to do anything." (P13)

Four participants reported a disconnection from their own bodies or a changed perception of their body parts, which fits into the depersonalization subcomponent of dissociation. Relevant excerpts follow:

"Do I feel lighter? Is my hand heavier or lighter? I can't tell if my head is farther or closer to the ground." (P09)

"It feels like there is a foam spray insulation around me, I feel like I can see my thoughts from a distant perspective." (P011)

"I felt disconnected from body." (P12)

"I feel like a 2D character." (P13)

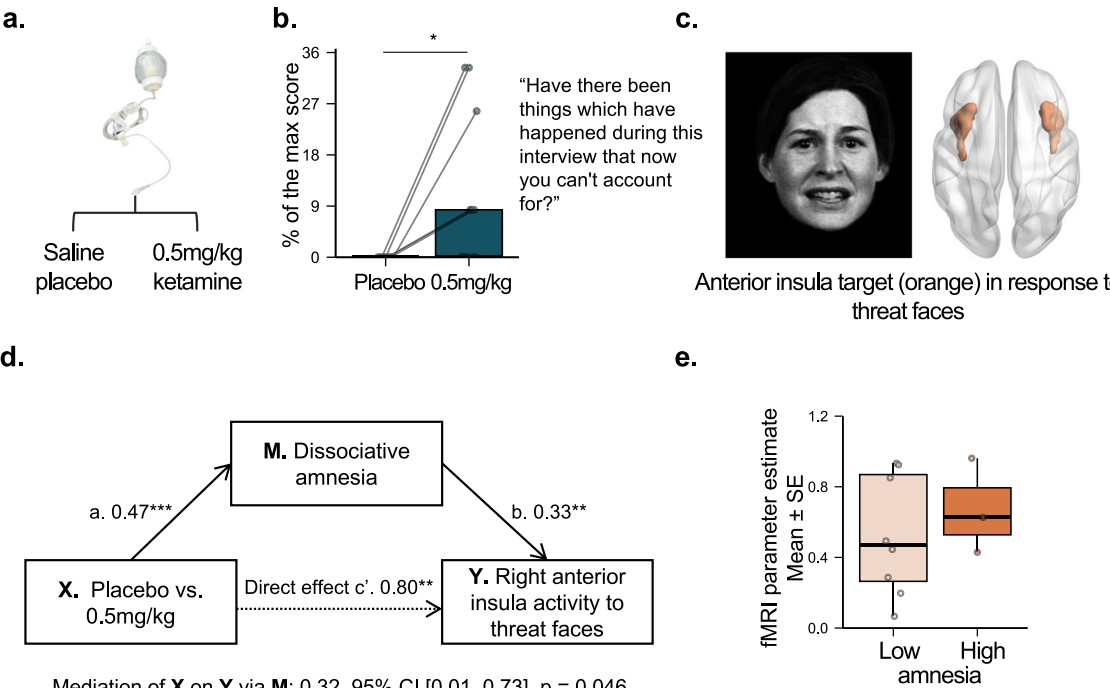

Fig. 2 | Dissociative experiences of amnesia mediate distinct ketamine-induced increases in insula activation in response to social threat. a Randomized dose (**X**) (*n* = 13 nonclinical participants). **b** Dissociative amnesia (**M**) at the end of infusion (40 min). Example items are listed. A two-sided post-hoc paired t-test was used to compare the effect of placebo vs. 0.5 mg/kg of ketamine on dissociative amnesia ($T_{12}$ = 2.50, *p* = 0.03, Cohen's *d* = 0.72, 95% CI = [1.16, 16.89]). We implemented an FDR correction to control for the testing of multiple scale sub-components. **c** Anterior insula engaged by threat faces measured by fMRI (**Y**). The facial expression was extracted from Fig. 1 of Williams et al.[47]. The image was modified with permission from the developers of a database of 3D facial stimuli described in Gur et al.[73]. **d** Ketamine-induced effects (**X**) on reducing right anterior insula activity

to threat faces (**Y**). An Averaged Causal Mediation Effect (ACME) mediation model was used to assess whether ketamine-induced dissociative amnesia mediates the effect of dose on acute changes in neural activity during emotional processing. No adjustments for multiple comparisons were made. **e** Right anterior insula activity in response to threat faces as a function of amnesia. In each boxplot, the central thick black line represents the median, color shaded boxes represent the first and third quartiles (the 25th and 75th percentiles), and the whiskers extend no further than 1.5 times of the distance between the first and third quartiles. *\*p* < 0.05, \*\**p* < 0.01, \*\*\**p* < 0.001. Source data are provided as a Source Data file. fMRI, functional magnetic resonance imaging. FDR, false discovery rate.

## Ketamine increases dissociative amnesia, anxiety, and impaired control and cognition in a dose-dependent manner

We observed a significant dose-dependent effect of ketamine in dissociative amnesia ($F_{2,76}$ = 5.46, *p* = 0.006, *pFDR* = 0.01; Fig. 2b for two conditions and Suppl. Fig. 1g for three conditions) with the significant increase only in the 0.5 mg/kg condition as compared to placebo ($T_{12}$ = 2.50, *p* = 0.03, Cohen's *d* = 0.72, 95% CI = [1.16, 16.89]). Two participants specifically mentioned "feeling spaced out" (P009, P012) during the infusion, related to the amnesia subcomponent of dissociation. We also found a significant dose-dependent effect in altered states of anxiety ($F_{2,25}$ = 28.58, *p* < 0.001, pFDR < 0.001; Suppl. Fig. 4b for two conditions and Suppl. Fig. 1h for three conditions) and impaired control and cognition ($F_{2,25}$ = 27.80, *p* < 0.001, *pFDR* < 0.001; Suppl. Fig. 1i) as measured by the 5D-ASC at the midpoint of the infusion (T = 20 mins). Post-hoc paired *t*-tests showed that 0.5 mg/kg dose versus placebo induced significant increase in dissociative amnesia ($T_{12}$ = 2.50, *p* = 0.03, Cohen's *d* = 0.72), anxiety ($T_{12}$ = 5.30, *p* < 0.001, Cohen's *d* = 1.53, 95% CI = [7.85, 18.82]), and impaired control and cognition ($T_{12}$ = 5.24, *p* < 0.001, Cohen's *d* = 1.51, 95% CI = [12.33, 29.87]) in the Suppl. Table 1.

## Ketamine increases threat-faces-evoked anterior insula and amygdala activity in a dose-dependent manner

We observed dose-dependent effects of ketamine on increasing right anterior insula ($F_{2,25}$ = 11.62, *p* < 0.001, the family-wise error corrected p-value (pFWE) < 0.05; Figs. 1c and 2c; Suppl. Figs. 1j and 4c) and right amygdala activity ($F_{2,23}$ = 10.93, *p* < 0.001, pFWE < 0.05; Suppl. Fig. 1k)

evoked by threat faces. Post-hoc t-tests (Suppl. Table 1) between the 0.5 mg/kg dose and the placebo revealed significant differences for the right anterior insula ($T_{10}$ = 3.22, *p* = 0.009, Cohen's *d* = 1.02, 95% CI = [0.21, 1.16]) and for the right amygdala ($T_{10}$ = 4.05, *p* = 0.002, Cohen's *d* = 1.28, 95% CI = [0.44, 1.50]). There were no significant ketamine dose-dependent changes in behavioral performance measures of accuracy (number of correct presses) and reaction time (Suppl. Methods and Results). Additional sensitivity analysis consistently showed the same dose-dependent effect of ketamine on anterior insula or amygdala activity when controlling for the effect of behavioral performance (Suppl. Methods and Results). We did not observe any effects of ketamine when examining anterior insula and amygdala activity as evoked by happy faces, or any effects of ketamine when examining sgACC and dACC as evoked by threat or happy faces.

## Ketamine-induced dissociative depersonalization relieves negative brain states assessed by anterior insula activity

We only tested for mediation effects comparing the 0.5 mg/kg and the placebo conditions, as the 0.05 mg/kg dose did not induce significant change in any mediators or in the neural activity of the right anterior insula or amygdala. Similarly, we only included activity in response to threat faces (and not happy faces) in mediation models, because only the response to threat faces showed a significant difference between the 0.5 mg/kg and the placebo condition. Our mediation analysis indicated that an increase in dissociative depersonalization negatively mediated or suppressed the effect of ketamine on task-evoked right insula activation in response to threat faces relative to neutral faces

(Fig. 1d). Specifically, ketamine at 0.5 mg/kg induced 0.39 SD decrease of insula activation through the mediation of increased depersonalization. Reflecting this suppressing effect, participants with high dissociative depersonalization by median split had a lower mean of ketamine-induced right anterior insula activity compared to participants with low dissociative depersonalization (Fig. 1e). The effect size of the mean difference was medium (Cohen's $d = 0.65$ although the two-sample t-test was not statistically significant ($T_9 = 0.96$, $p = 0.37$, 95% CI = [−0.24, 0.63]). We did not find any mediation relationships for other ASCs hypothesized to relieve negative affective brain states, including dissociative derealization or a blissful state.

### Ketamine-induced dissociative amnesia exacerbates negative brain states assessed by anterior insula activity

In contrast to depersonalization, ASCs hypothesized to promote negative affective brain states, specifically dissociative amnesia (Fig. 2d), positively mediated the effect of ketamine on right anterior insula activation in response to threat faces relative to neutral faces. Specifically, ketamine at 0.5 mg/kg induced 0.32 SD increase of insula activation through the mediation of increased dissociative amnesia. Reflecting this mediating relationship, participants with high dissociative amnesia by median split had a higher mean of ketamine-induced right anterior insula activity compared to participants with low dissociative amnesia (Fig. 2e). The effect size of the mean difference was medium (Cohen's $d = −0.69$) although the two-sample t-test was not statistically significant ($T_9 = −0.75$, $p = 0.48$, 95% CI = [−0.66, 0.37]). We did not observe a significant mediation effect for anxiety or impaired control and cognition, but the mediation effect for anxiety was marginally significant in the same direction as dissociative amnesia ($p = 0.08$, Suppl. Fig. 4). No mediation relationships were observed for the amygdala. A summary of all mediation analyses is present in Suppl. Table 3.

## Discussion

The present findings advance our understanding of the role of ketamine-induced ASCs in mediating neural changes in the brain's affective circuitry. In our placebo-controlled, within-participant design which examined nonclinical participants, we conducted multi-modal measurements of dissociation and other ASCs as well as emotional-faces-evoked affective circuit activity before, during, and/or within ~2 h of racemic intravenous (IV) ketamine (0.05 mg/kg or 0.5 mg/kg) or placebo infusion. We found that ketamine-induced acute, dose-dependent increases in ASCs—including dissociative depersonalization, derealization, and a blissful state—which we hypothesized would alleviate negative affective states. Ketamine also induced dose-dependent increases in acute ASCs associated with negative affect, including dissociative amnesia, anxiety, and impaired control and cognition. There was no effect of placebo on these measures. These findings align with previous work in the mechanistic literature that showed significant increases in dissociation in response to ketamine (e.g., refs. 28,33,34), which demonstrates the rigor of our placebo-controlled, within-participant design.

A key insight from this trial is that the acute effect of sub-anesthetic ketamine on the brain's negative affect circuitry may be specific to and dependent on the level of specific aspects of dissociation and other ASCs. We measured reactivity within affective circuitry using a well-established facial emotion task, focusing on the anterior insula, amygdala, and ACC. We found that compared to lower dissociative depersonalization, higher dissociative depersonalization mediated reduced right anterior insula activity in response to negative emotional stimuli, whereas higher dissociative amnesia mediated an increase in task-evoked right anterior insula reactivity. These findings suggest that at least two mechanisms of action contribute to the acute dissociative effects of ketamine on the brain's affective circuitry. Specifically, induction of the depersonalization aspect of dissociation

might be an essential ingredient in the mechanisms by which ketamine alleviates negative affective brain states. Ketamine's effects on depersonalization, or the sensation of detaching from one's body, might be accompanied by detachment from negative affective states of emotional pain including depression[11]. This suggestion draws on findings in PTSD in which dissociation is understood to allow for detachment from negative emotional states[35] and pain perception[36]. Dissociation in PTSD is also associated with enhanced insula and amygdala reactivity during fMRI of the same nonconscious social threat stimuli as used in the present study[37]. On the other hand, induction of dissociative amnesia and anxiety may contribute to the mechanisms by which ketamine can increase negative affective brain states. The overall level of these different aspects of dissociation and other ASCs induced by ketamine is likely important in its effect on affective brain states.

Our mediation findings might help to account for potentially inconsistent findings across studies, methodologies, and healthy versus clinical participants. When we considered the group effects in our trial, without considering variability in different aspects of dissociation, ketamine induced an increase in the average threat-faces-evoked reactivity of the right anterior insula and right amygdala compared to placebo. The emotion task deployed in our trial is designed to evoke reactivity of these regions, including under drug conditions. Prior trials imaging facial emotion-evoked activity after two days post-ketamine have observed increases in amygdala for individuals with MDD, but decreases in the insula, and opposing patterns of change for healthy individuals[28,34]. In studies deploying imaging immediately after an IV infusion, healthy individuals also show reductions in acute ketamine-induced insula and amygdala reactivity, compared to a non-IV baseline, while performing a task that assesses the effects of emotional stimuli on a cognitive process, verbal working memory[26,27]. There are several possible explanations for these variations in findings. The direction of effect might differ with both clinical status of participants and the duration of the post-infusion period and we address this further under potential study limitations. In the present study, the use of a randomized placebo comparison controlled for the likely arousing and contextual effects of the IV line, which by itself may increase limbic reactivity. Our randomized placebo control also addressed the likelihood that findings may reflect the natural attenuation of neural reactivity when using a non-randomized, non-IV baseline, as was used in these prior designs[26,27].

Let's examine the translational therapeutic implications of this study's results. Dissociative effects induced by ketamine are known to commonly occur at therapeutic doses for TRD in over 70% of participants with IV ketamine[38] and 27% of participants with esketamine[39]. Although dissociation is generally viewed as an adverse effect in the medical model of ketamine administration[16], there remains debate about whether and how dissociation and other altered states are involved in ketamine's antidepressant effects[15]. Some blinded studies of both single[40,41] and multiple[42] infusions have found no relationship between intra-infusion dissociation and antidepressant response, whereas other studies of both single[18,43–45] and multiple[17] infusions have found correlations. A possible explanation for the mixed findings may be differences in the relative subcomponents of dissociation experienced by individuals in the studies. Indeed, a study of a single ketamine infusion found that the depersonalization subscale of the CADSS was positively associated with the change in depression symptoms over multiple time points (from 230 min to 7 days after infusion), and derealization was positively associated with the change in depression symptoms at Day 7[17]. Another trial of a single ketamine infusion showed that altered states involving the experience of spirituality, unity, and insight from the 5D-ASC rating scale in particular were associated with improvement in depression scores[18]. In another study, anxiety as measured by the 5D-ASC during a first ketamine infusion was predictive of non-response in depressed participants after six infusions[19]. These findings speak to the importance of separately

assessing different components of dissociation and other ASCs, as we have done in this study.

We found that the altered states of depersonalization and amnesia significantly mediated the effect of a therapeutic antidepressant dose of racemic IV ketamine (0.5 mg/kg) on threat-faces-evoked right anterior insula reactivity, while anxiety showed a marginal mediation in the same direction as the amnesia. The states have been previously found to relate to longer-term antidepressant response. Depersonalization assessed by the CADSS, but not overall dissociation, has been found to predict antidepressant response to racemic IV ketamine[17], whereas anxiety assessed by the 5D-ASC predicts non-response in depressed participants[19]. If the current results in nonclinical participants can be extrapolated to individuals with depression, they may shed light on the mechanisms by which these specific dissociative states may predict response or non-response to ketamine. Intra-infusion depersonalization may drive an acute reduction in insula reactivity to negative emotion and, thus, a reduction in negative affective states capable of translating into a long-term antidepressant effect. Indeed, therapeutic studies of IV ketamine in depressed patients show that during the window of time that dissociation occurs, significant reductions in depressive symptoms are observed[12–14]. Furthermore, these reductions are maintained for up to 2 weeks after an infusion[12–14], and an early decrease in depressive symptoms predicts improved depression outcomes at the end of the treatment course[19,42]. Thus, acute changes in anterior insula activity in response to negative emotional stimuli may be an early biomarker of longer-term activity changes in this region. Successful antidepressant treatment lowers insula reactivity to negative stimuli[46], and our prior work using the same Facial Expressions of Emotion Task (FEET) of the current study in a behavior intervention study has found that early change in insula reactivity to negative stimuli predicts future treatment outcomes as a function of treatment[21] Another prior pharmacotherapy study also using the same FEET has shown that reactivity of another critical affective brain region—the amygdala—to negative emotional stimuli is predictive of antidepressant response[47]. Additionally, a study of depression that included qualitative assessments revealed that participants felt that a longer-term decrease in suicidal ideation was related to acute dissociative effects during infusion[14].

While the dissociative effects of ketamine have been most studied in the context of depression, our findings may have implications for other disorders for which ketamine is being studied and insula reactivity has been shown to play a key role, including SUDs. A qualitative study of experiences while undergoing three ketamine infusions for treating alcohol use disorder found that participants reported that dissociation was important to changing their relationship with alcohol and that the infusion experience decreased their motivation to drink to relieve negative emotional states[48]. Furthermore, there is evidence that the insula may be involved in all three stages of addiction[29,49], including the withdrawal/negative affect stage. Our findings suggest that one mechanism by which ketamine may effectively treat SUDs involves detachment from the negative affect that is related to withdrawal.

These findings may also be relevant for identifying who will respond to ketamine within the first session. Response and remission rates for IV ketamine for depression are 53.6% and 28.9% in a real-world sample, and some individuals have worsening of depression (8%) and suicidal ideation (6%) with ketamine treatment[50]. Thus, a key clinical need is to identify a priori, or after a single infusion, who is most likely to respond to ketamine in order to avoid multiple exposures to a drug with short-term[16] and potentially long-term[51–53] adverse effects as well as worsening of depression symptoms. Our results suggest the possibility that those with higher intra-infusion depersonalization and lower dissociative amnesia and altered states of anxiety during a first infusion may have a better long-term response given the reduced anterior insula response in these participants. The current study necessarily restricted the range of symptom features due to it being designed as a

healthy mechanistic study, but findings to date indicate the need for future multi-modal studies that expand the range of symptom features and include participants with relevant psychiatric diagnoses.

Let's examine the reverse translational implications of the current study's results. Future reverse translational work in mouse models will be essential for developing a better understanding of the causal relationships between dissociation as measured by behavioral paradigms, such as the one developed by Deisseroth and colleagues[54], and anterior insula reactivity to negative emotional stimuli. This same research group previously found that ketamine induced an oscillatory rhythm in layer 5 neurons of the retrosplenial cortex (RSP), which is necessary for dissociation-like behavioral effects[54]. Although the researchers completed an unbiased pan-cortical assessment of the mouse brain to identify the neural origins of ketamine-induced dissociation, they were not able to fully assess deeper regions of the brain including the insula. It would be interesting to examine the mouse insula after administration of ketamine to determine whether a similar oscillatory rhythm is present as was found in the RSP. Interestingly, while the mouse RSP is not structurally connected to the insula[55], both rodent[56] and human[57,58] studies have demonstrated a functional anti-correlation between activity in the RSP/posterior cingulate cortex and the insula.

The insights derived from this study must be considered within the context of the study's limitations. First, because our focus was on multi-measurement, multi-domain assessment of ketamine's acute effects, the sample size was limited and not intended to provide a basis for making broader group average inferences. It will be essential to conduct studies designed to replicate and expand these findings. Second, our multi-domain methodology necessarily relies on the administration of a series of assessments at multiple time points within each IV session. While we made every effort to administer assessments within the exact same timing for each participant, this was not always possible due to the need to first address side effects such as nausea. The third limitation relates to the temporal sequence of our mediation models, which was selected based on the order of our measures. Clearly, a change in brain state must have occurred in order for dissociation and other ASCs to occur. It is not clear from our current design whether this brain state is the same as the one we measured during our functional neuroimaging session. Thus, we cannot conclude that depersonalization and dissociative amnesia caused the differential changes in task-evoked anterior insula that we observed. This question may be examined in designs that use more frequent repeated sampling with higher temporal-resolution neuroimaging tools. Fourth, because ketamine has an impact across multiple biological systems, we recognize that there might be additional mediators on the activation of the insula in response to threat, including physiological measures such as heart rate and blood pressures[59]. Although we have shown that physiological measures of heart rate and blood pressures are not significant mediators of the changes in insula activity in our sample, future studies are needed to test additional potential measures, including ketamine metabolites and resting-state magnetoencephalography (MEG) gamma power[60]. A fifth limitation relates to the extent to which findings from healthy volunteers generalize to an understanding about antidepressant response to ketamine. Previous functional neuroimaging studies observe inconsistent neural effects of ketamine across groups of healthy individuals and patients with MDD. Both groups have been found to show a similar emotional valence-related of effect of ketamine versus placebo on ACC subregion activity elicited by anger versus happy in a facial emotion task, along with opposing effects on another ACC subregion and frontal cortex for post-ketamine versus post-placebo[28]. Healthy individuals and patients with MDD have been found to show opposing effects on activity in the temporal gyri and precuneus/posterior cingulate during an implicit emotion task following ketamine versus placebo[34]. In the resting state, both groups have been observed similar effects of ketamine on gamma power in the ACC and insula[61] while patients with MDD specifically have been found

to show a normalization of insula to default mode connectivity[62]. These previous studies have examined neural changes two days after ketamine infusion and were not designed to assess the acute neural effects immediately following ketamine infusion as in the present study. However, the variability of findings across healthy and MDD groups highlights the need for future investigations that expand upon the present findings to include clinical participants, enabling a direct comparison of the acute neural effects of ketamine in healthy controls with the acute antidepressant response. Such future investigations would also be important for understanding the relationship of dissociation, assessed by the CADSS and other measures, to the clinical antidepressant response, given that dissociative experiences can also be more intense or extended in MDD[61].

To our knowledge, this work provides an initial demonstration that acute ketamine-induced changes in anterior insula reactivity to negative emotional stimuli depend upon specific aspects of dissociative experiences and other ASCs. These findings shed light on the neurobiological mechanisms that underlie ketamine's ability to both acutely relieve negative affective states and induce them in nonclinical participants. Furthermore, if these results are extended to patients with depression, they should advance scientific understanding about how aspects of intra-infusion ASCs might predict depression response −results that might help inform more personalized treatment interventions.

## Methods
### Priori power analysis
To estimate the sample size for detecting ketamine's effect on our primary neural dependent measures of negative affect circuit activation in response to emotional stimuli, we drew on prior reported effect sizes for a prior study in which imaging was similarly undertaken immediately following ketamine infusion in a repeated measures design[27]. In this prior study the effect size reported for left and right amygdala activity was $\eta^2 = 0.43$ and $\eta^2 = 0.31$ respectively. For primary self-reported dissociation effects of interest, we similarly drew on prior reported effect sizes in a separate study[61], and the effect size was $\eta^2 = 0.38$. Using the most conservative effect size of $\eta^2 = 0.31$, which was converted to the Cohen's $f$ of 0.67, we conducted a power analysis using G*Power Version 3.1.9.6[63] for sample size estimation. The sample size needed to detect an effect on dissociation or neural activity with $\alpha = .05$ and at least 95% power for a within-subject design with two repeated imaging measurements is $n = 10$. Here we overestimated the needed sample size by including only two dose conditions (the placebo and the ketamine at 0.5 mg/kg) to match previous study designs. Still, the obtained sample size of $n = 13$ is adequate to detect ketamine's dose-dependent effects on primary measures of interest.

### Participants
We recruited from the community 13 nonclinical adult participants aged 18 to 55 years (mean = 33 years, SD = 9.82 years), with an equivalent distribution of self-reported biological sex (female: 54%, male: 46%). The first participant was enrolled on August 19, 2019 and the last participant was enrolled on October 21, 2022. Data was collected using REDCap versions 9.2.5 through 12.5.13. All participants passed the study screening procedure, reported ≥2 prior uses of ketamine per IRB requirements (Suppl. Methods), and endorsed minimal clinical symptoms at baseline consistent with inclusion and exclusion criteria (For a full list of eligibility criteria, see Suppl. Table 4). For detailed demographic and clinical symptom information, see Suppl. Table 5. Participants provided written informed consent, and the protocol was approved by the Stanford Institutional Review Board. The study was preregistered on clinicaltrials.gov under the identifier NCT03475277 on March 23, 2018. Information pertaining to data availability can be found in the "Data Availability" statement. The

internal study protocol is available upon reasonable request to the corresponding author.

### Visits
In our randomized, cross-over design, participants underwent a total of five visits, including the screening visit, baseline visit, and three infusion visits during which the participants, research coordinator (study assessments), research nurses (peripheral IV catheter placement), and licensed study clinicians (ketamine infusions, monitoring, and safety assessments) were blinded to the intervention. At any of the three infusion visits, participants received placebo saline, 0.05 mg/kg ketamine, or 0.5 mg/kg ketamine with the order randomized for each participant. We used IV ketamine administration as it provides the most predictable dosing with 100% bioavailability. Given the within-participants design of the study, each participant received all three of the specified doses across the duration of the trial. Each infusion visit was separated from any other infusion visit by 10–14 days to avoid drug carry-over effects. Participants arrived fasted for infusion visits in the morning to reduce the risk of emesis, and they completed a urine drug screen and pregnancy test (if applicable), had their baseline vitals recorded, and had a peripheral IV catheter placed at the Clinical Trial Research Unit at Stanford University. Participants were then escorted to the Stanford Center for Cognitive and Neurobiological Imaging, where they were met by the study clinician and subsequently began the infusion.

Racemic IV ketamine (0.05 mg/kg or 0.5 mg/kg) or 0.9% normal saline (Mariner Advanced Pharmacy Corp, San Mateo, CA) was delivered via an elastomeric pump (Braun Easypump® ST/LT) over a period of 40 min. For safety reasons, pulse, blood pressure, and pulse-oximetry were monitored and recorded every 10–15 min for all conditions during the infusions and scanning sessions. The total visit time for infusion visits was 6–8 h.

### Assessments of dissociation and other ASCs
To address our first objective to test whether specific aspects of dissociation and other altered states of consciousness differed across dose conditions, we utilized the CADSS and the 5D-ASC to assess specific aspects of ketamine-induced dissociation and other ASCs. CADSS is a 23-item clinician-administered questionnaire that was assessed pre- ($T = 0$ min) and post-infusion ($T = 40$ mins) by a study clinician. CADSS includes three subscales: "Depersonalization", "Derealization", and "Amnesia". While the CADSS is commonly used to assess ketamine-induced dissociative states, some evidence suggests that it may not fully capture the acute ketamine experience[64]. Thus, we also utilized the 5D-ASC, a 94-item scale that evaluates drug-induced change in subjective experience or psychological functioning compared to normal waking consciousness. The 5D-ASC can be divided into five validated dimensions ("Oceanic Boundlessness", "Dread of Ego Dissolution", "Visionary Restructuralization", "Auditory Alterations", and "Vigilance Reduction") and 11 lower-order subscales derived from 42 items using a data-driven approach[65]. Among the 11 subscales, the "Blissful State" subscale includes items about experiencing pleasure, peace, and love; the "Anxiety" subscale involves items surrounding fear of being in an altered state and perceiving the environment as changed; and the "Impaired Control and Cognition" subscale includes items about not being able to complete a thought and the feeling of no longer having a will of one's own.

We chose to assess blissful state, anxiety, and impaired control and cognition from the 5D-ASC to complement the CADSS, since these either directly assess an affective state or are likely to induce a particular affective state. Participants completed the 5D-ASC scale after scanning was complete and were instructed to rate retrospectively as if they were 20 min into the infusion. Building upon previous evidence of ASCs predicting treatment responses, in this report we included dissociative depersonalization, derealization, and a blissful state as ASCs

that we hypothesized would relieve negative affective brain states or promote positive affective ones, while dissociative amnesia, anxiety, and impaired control and cognition were hypothesized to exacerbate negative affective brain states or dampen positive affective ones. In addition to the above quantitative assessments, we used keywords from the CADSS and 5D-ASC to prompt participants to describe their experience, and we recorded narratives for the last five participants (P09–P13).

### Linear mixed effects models and t-test analyses of dose-dependent effects of ketamine on dissociation and other ASCs

To examine dose-dependent effects of ketamine on CADSS-assessed subcomponents of dissociation and 5D-ASC-assessed other ASCs, we used linear mixed effects models (LMMs) with dose (placebo, 0.05 mg/kg or 0.5 mg/kg) as the fixed effect and participant as a random effect using the lmer package (https://cran.r-project.org/web/packages/lme4/index.html) in R version 4.0.5 (https://www.r-project.org/). Time and dose-by-time interaction were added if applicable (Suppl. Methods). Age and biological sex were included as covariates, although we were under powered to draw any conclusion with the current sample size. We implemented an FDR correction to control for the testing of multiple scale sub-components. For significant dose-dependent effects, post-hoc paired t-tests were also run to compare 0.5 mg/kg versus placebo, 0.05 mg/kg versus placebo, and 0.5 mg/kg versus 0.05 mg/kg, to reveal which drug dose condition drove the effect. Effect size of Cohen's $d$ was calculated for paired t-tests using the effectsize package (https://cran.r-project.org/web/packages/effectsize/index.html).

### Acquisition of brain fMRI during a Facial Expressions of Emotion Task (FEET)

Upon completion of the 40-min infusion, participants were transferred to the MRI suite within the Stanford Center for Cognitive and Neurobiological Imaging via a wheelchair to begin scanning at approximately 60 min after initiation of infusion. FEET was collected at approximately 2 hours after initiation of infusion. Imaging data were acquired on a GE Discovery MR750 3T scanner using a Nova Medical 32-channel head coil at baseline and at the three drug visits.

**Non-conscious FEET fMRI.** At all three drug visits, participants non-consciously viewed facial expressions of emotion, during which fMRI data were collected, to probe automatic bottom-up activation of the positive and negative affect circuits[66] (see Suppl. Fig. 2). Stimuli were from a standardized series of facial expressions of reward-related emotions (happy), threat-related emotions (fear, anger), and loss related emotions (sad), along with neutral expressions. Stimuli were modified such that the eyes were presented in the central position of the image. Participants were instructed to press a button when seeing a face, and we controlled for active attention by monitoring alertness with an eye tracking system. Each face was presented for 16.7 ms (1 monitor frame), followed immediately by a neutral face perceptual mask for 150.3 ms (9 monitor frames) and an inter-stimulus interval of 1083 ms. Neutral masked faces stimuli were offset slightly by 1° in random directions to control for the possible detection of emotions based on perceptual features (e.g., the apparent motion in the pairing of a fear face with upraised eyebrows followed by a neutral face mask compared with the pairing of an angry face with contracted eyebrows followed by a neutral face mask). Using behavioral psychophysiological testing, we have shown that when faces in this paradigm are presented at ≤20 ms, they meet signal detection criteria for being at the subliminal threshold for detection such that individual participants cannot consciously detect the presence of the face nor discriminate the facial expression[67]. A total of 240 masked emotional faces from 6 different emotions were presented, with 8 faces of the same emotion forming a block and each emotion block repeated for 5 times in a pseudorandom order. BOLD fMRI was acquired with axial slices in an interleaved order, with TR = 2 s, TE = 27.5 ms, FA = 77°, acquisition time = 5:08, FOV = 222 ×222 mm, voxel size = 3 mm isotropic. One hundred fifty-four volumes were acquired, with the first five volumes cut to account for the non-steady state. See the Suppl. Methods for details of the quality control procedure and imaging preprocessing steps.

### Analysis of brain FEET fMRI data

**Generating activation maps.** Preprocessed data were entered into a general linear model at the individual level using SPM8 (https://www.fil.ion.ucl.ac.uk/spm/software/spm8/). Each block of emotional expressions was convolved with a canonical hemodynamic response function, and the blocks were used as regressors in the general linear model, as were motion spikes. Activation maps for threat (fear and anger facial expressions) relative to neutral faces, and for happy relative to neutral faces, were estimated to examine ketamine-induced brain activity change in response to negative and positive emotions. We use the term positive affective brain states to mean neural activity that is evoked by positively valenced emotional stimuli, and that may be associated with subjective positive experiences, whereas we use the term negative affective brain states to refer to neural activity that is evoked by negative emotional stimuli such as social threat, and that may be linked with subjective negative experiences such as anxiety.

**FEET neuroimaging data analysis of dose-dependent effects of ketamine on affective neural circuit.** To examine our second objective to test the dose-dependent effects of ketamine on brain activity in response to emotional expressions, we conducted a one-way repeated Analysis of Variance in SPM—with dose as the within-participant factor—on the activation maps for threat faces (consisting of both anger and fear faces) relative to neutral faces, and happy faces relative to neutral faces. Based on the pre-specified primary focus of anterior insula, amygdala, and ACC neural targets, we constrained our voxel-wise analysis using masks consisting of bilateral anterior insula, amygdala, and ACC. Conducting within-region voxel-wise analyses instead of deriving an average value per region of interest (ROI) enabled us to focus on the ROIs while still obtaining precision in detecting which part within the region is showing an effect. The definition for ROIs of anterior insula, amygdala, and ACC was established in our previous work[20] (Suppl. Methods and Suppl. Figs. 1k and 2i for anterior insula and amygdala). To correct for multiple comparisons, a voxel threshold of $p < 0.001$ and a Gaussian random field theory (GRF) family-wise error (FWE) cluster-level correction at $p < 0.05$ was applied. For clusters that survived multiple comparison corrections, we extracted the peak voxel activation (fMRI beta estimate) for all three conditions and conducted planned contrasts using paired t-tests between each pair of dose conditions and reported the Cohen's $d$ as the effect size, as we did with the ASC data. Behavior performances of the FEET task were also extracted and tested for ketamine's dose-dependent effects (Suppl. Methods).

**Mediation analysis using an averaged causal mediation effect model.** To address our third objective—to test whether specific aspects of ketamine-induced dissociation and other ASCs mediate the effect of dose on acute changes in neural activity during emotional processing—we utilized the Averaged Causal Mediation Effect (ACME) mediation model[68, 69]. ACME models were used to test our working hypotheses that: 1) ketamine will reduce neural activity reflecting relief of negative affective states, mediated by depersonalization and derealization from the CADSS and blissful state from the 5D-ASC; and 2) ketamine will increase neural activity reflecting exacerbation of negative affective states, mediated by dissociative amnesia from the CADSS, as well as anxiety and impaired control and cognition from the 5D-ASC. ACME is

a method for dissecting the total effect of an intervention (in this case ketamine) into direct effects on neural activity and indirect effects on neural activity that are transmitted via a mediator (in this case dissociation or other altered state of consciousness). It addresses the limitation of analysis designs that incorrectly treat intermediate or mediator variables as confounding factors. In our design using ACME, the intervention is designated as the independent variable 'X' (0.5 mg/kg versus placebo), dissociation or other altered state of consciousness is the mediator variable 'M' and emotion task-elicited neural activity is the dependent variable 'Y'. We implemented the ACME using the mediation package (https://cran.r-project.org/web/packages/mediation/index.html) in R version 4.2.3, in combination with the lmer package to account for the repeated design. The X variable was coded as 0 (placebo) and 1 (0.5 mg/kg), and the M and Y variables were standardized. This approach enabled us to estimate effect size for the average direct effects of X on Y, indirect effects of X on Y transmitted via M and the total effect for the overall ACME model[70]. The primary effect size of interest is the indirect mediation effect, represented as ketamine-induced change in neural activity in the magnitude of the standard deviation transmitted by the change in ketamine induced dissociation or altered state of consciousness. Because these effect sizes were standardized they can be compared across studies and interpreted similarly to correlations[71], <0.2 is considered a weak effect; ≥0.2 and ≤0.5, a moderate effect; and >0.5, a strong effect. Statistical significance ($p$ value) was generated using bootstrapping procedures with 1000 resamples. As an exploratory step, we also examined the intra-subject correlations between M measures separately (Suppl. Methods).

### Reporting summary

Further information on research design is available in the Nature Portfolio Reporting Summary linked to this article.

## Data availability

The pre-processed fMRI and questionnaire data are available under restricted access because they contain data acquired to address aims independent from those reported in the present manuscript and thus have not been fully analyzed yet. Access can be obtained on reasonable request by contacting the corresponding author. The raw fMRI data are protected and are not available due to data privacy laws. The data generated in this study to produce figures and tables are provided in the Supplementary Information/Source Data file and are available in the GitHub database at https://github.com/WilliamsPanLab/Ketamine-FEET-Mediation[72]. Source data are provided with this paper.

## Code availability

The full analysis codes are available at https://github.com/WilliamsPanLab/Ketamine-FEET-Mediation[72]. Simulated data are provided to test run all group level analyses.

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

## Acknowledgements
This work was supported by the National Institute of Drug Abuse under award P50DA042012 (Overall PI: K.D., Project 4 PI Project Lead: L.M.W.). The sponsor had no role in the study design, data collection and analysis, or manuscript writing. This work was also supported by Stanford's Clinical and Translational Science Award Program overseen by the National Center for Advancing Translational Sciences at the National Institutes of Health (UL1TR003142-01). We acknowledge the participants and the editorial support of Jon Kilner, MS, MA (Pittsburgh, Professional Medical & Scientific Writing/Editing, USA), who received compensation for his role. We would also like to thank Investigational Drug Pharmacist Henry Truong, PharmD & team (Mariner Advanced Pharmacy, San Mateo, CA) for procuring, compounding, and dispensing the ketamine solution and its matching placebo in a randomized order as well as providing educational materials on the elastomeric pump.

## Author contributions
L.M.W., B.D.H., B.K. and K.D. conceptualized and designed the study. L.M.H., X.Z. and L.M.W. wrote the manuscript. X.Z. performed the analyses. L.M.H., B.D.H, P.J.vR. and R.H. monitored the ketamine infusions as licensed clinicians. N.J.G. and C.B. acquired the data. T.S., P.J.vR., J.A.Y., C.I.R., B.K. and K.D. critically reviewed the manuscript. All authors participated in the discussion and revision of the manuscript and approved the final manuscript. L.M.W. supervised the overall project and L.M.H. supervised the clinical components of the project.

## Competing interests
L.M.H. in the past 3 years has served on an advisory board for Roche. T.S. in the past 3 years has reported grants from Pathway Genomics, Stanley Medical Research Institute, Elan Pharma International Limited, Merck and Co., and Sunovion Pharmaceuticals; consulting fees from Allergan, Impel NeuroPharma Inc., Intra-Cellular Therapies, and Sunovion Pharmaceuticals; honoraria from CME Institute, Health and Wellness Partners Inc., and CMEology; royalties from Wolters Kluwer Health (UpToDate), Jones and Bartlett, American Psychiatric Association Press, and Hogrefe Publishing. C.R. in the last 3 years, has served as a consultant for Epiodyne and received research grant support from Biohaven Pharmaceuticals and a stipend from APA Publishing for her role as Deputy Editor at The American Journal of Psychiatry. L.M.W. has served as a scientific advisor for One Mind PsyberGuide, a member of the executive advisory board for the Laureate Institute for Brain Research and holds patent 16921388 (Systems and Methods for Detecting Complex Networks in MRI Image Data) unrelated to the present study. The remaining authors declare no competing interests.

## Additional information

[1]Department of Psychiatry and Behavioral Sciences, Stanford University School of Medicine, Stanford, CA, USA. [2]Sierra-Pacific Mental Illness Research, Education and Clinical Center (MIRECC), Veterans Affairs Palo Alto Health Care System, Palo Alto, CA, USA. [3]Department of Anesthesiology, Perioperative and Pain Medicine, Stanford University School of Medicine, Stanford, CA, USA. [4]Veterans Affairs Palo Alto Health Care System, Palo Alto, CA, USA. [5]Department of Bioengineering, Stanford University, Stanford, CA, USA. [6]Howard Hughes Medical Institute, Stanford University, Stanford, CA, USA. [7]Department of Psychology, Stanford University, Stanford, CA, USA. [8]These authors contributed equally: Laura M. Hack, Xue Zhang. [9]These authors jointly supervised this work: Carolyn I. Rodriguez, Brian Knutson, Leanne M. Williams. ✉e-mail: leawilliams@stanford.edu

