## [Peer Review File · Nature Communications]

REVIEWER COMMENTS

Reviewer #1 (Remarks to the Author):

This is an interesting analysis of 13 healthy volunteers who underwent ketamine infusion and MRI scanning. The authors link dissociative symptoms induced by ketamine as defined by clusters from the CADSS and ASCs with insula and amygdala activity in response to a face processing task. Strengths include the novel research design, strong methods and the use of multiple ketamine doses. Limitations include the small sample size and the generalizability of findings from a healthy volunteer sample to understand the antidepressant response to ketamine. Additionally, as the authors state, the causal mechanisms of dissociation on functional neuroimaging results cannot be determined.

1.) My primary concern is the extension of these results to the antidepressant effects of ketamine. Work by Nugent (2018), Reed (2018, 2019) and Evans (2018) have shown diverging brain responses to ketamine in healthy volunteers, using both resting state and emotional tasks. The relationship of antidepressant response to CADSS experience is also divergent as healthy volunteers often show a transient dysphoric effect of ketamine. Please include additional discussion of the limitations of using healthy volunteer samples to draw conclusions about antidepressant response in depressed patients, using this literature.

2.) My other concern is the “meaning” of the dissociative experience as either causal or an epiphenomenon. Ketamine has an impact across multiple biological systems, so it’s difficult to know whether correlated changes in heart rate, ketamine metabolites or gamma power would have had a similar effect on amygdala and brain activity. The factor scores developed from these dissociative scales are so interrelated that it is difficult to parse whether one symptom has a distinct effect from the others. Please include more discussion of why dissociation may be distinctly related to emotional changes after ketamine, as compared to other potential biomarkers.

Reviewer #2 (Remarks to the Author):

This manuscript by Hack et al. describes the findings of a study using mediation analyses to investigate the effects of ketamine-induced altered states of consciousness on brain activation in two regions (bilateral Anterior Insula and Amygdala) involved in the processing of affective material. In a double-blind within-subject design a total of 13 healthy adults received three temporally separate infusions of saline, 0.05mg/kg and 0.5mg/kg ketamine in a random order. Before and after the infusion the CADSS was administered to assess dissociative symptoms followed by an fMRI scan to probe neural responses to implicit affective faces by means of the FEET. Subsequently, the 5D-ASC was employed retrospectively to assess blissful state, anxiety and impaired control and cognition. A linear mixed effects model was

used to test the respective hypotheses. Mediation analysis showed that more pronounced dissociative depersonalization attenuates the activity of the right anterior insula in response to threat faces. Furthermore, dissociative amnesia and anxiety (trend) was found to amplify the effect of ketamine on anterior insula activity in response to fearful faces. However, no significant results were shown for the mediating effect of ASCs on amygdala reactivity.

Major points:

- Please report descriptive statistics and effect sizes for all statistical tests conducted.
- Please also report the statistics of all non-significant findings.
- In the introduction you stated that detachment, as a component of dissociation, may exert an ameliorative effect on negative affective and pain states. Did you find evidence for that assumption beyond the antidepressant effects of ketamine?
- Please define “positive and negative (brain) states” more clearly.
- Please report behavioral data (button presses per condition) from the FEET and add analyses to control that ketamine did not change the subliminal thresholds.

Minor points:

- The inclusion criteria require participants to have used ketamine at least two times previously to the administration in the study. What are the reasons for defining this criterium?
- Please move all quantitative ASC to the supplement.
- Did you control for potential order effects of the 3 conditions in your within-subjects design?
- What kind of coefficients are depicted in Fig. 1.d., Fig. 2.d. and Fig. 3.d.?
- Please indicate more clearly in Fig. 1,2 and 3 that findings only apply for the RIGHT anterior insula and for threat faces only
- Please improve the understanding of the following sentences: “ Because the n-back emotion task relies on cognitive regulation/suppression of limbic reactivity, it is also possible that less participant-level dissociation, relevant to the esketamine formulation, is required for attenuation of this reactivity.” (p. 13) and “Indeed, successful antidepressant treatment lowers insula reactivity to negative stimuli³⁸, and our prior work using the same FEET task of current study in a novel behavior intervention study has found that early change in insula reactivity to negative stimuli predicts future treatment outcomes as a function of treatment ³⁹, and another prior pharmacotherapy study also using the same FEET has shown that reactivity of another critical affective brain region – the amygdala – to negative emotional stimuli is predictive of antidepressant response⁴⁰.” (p.14).

Reviewer #3 (Remarks to the Author):

The article discusses a study on the effects of ketamine on the brain's affective circuitry. The study found that ketamine induced both positive and negative altered states of consciousness (ASCs) and that the specific type of ASC induced by ketamine may be crucial to its antidepressant effects. The study's findings suggest that previous mixed results on the relationship between dissociation and antidepressant response may be due to differences in the subcomponents of dissociation experienced.

Overall, the study provides important insights into the complex relationship between ketamine-induced ASCs and neural changes in the brain's affective circuitry. The findings suggest that the effects of ketamine on negative affective states are dependent on the specific aspects of ASCs induced by the drug. This information could be useful in developing more targeted and effective treatments for depression and other mood disorders.

While the study provides valuable insights into the effects of ketamine on altered states of consciousness and its potential therapeutic use, there are some concerns that need to be addressed. Firstly, the sample size of nonclinical participants used in the study is relatively small, which may limit the generalizability of the results to clinical populations. A power analysis could be provided to support the adequacy of the sample size. Secondly, the authors should be more precise in the statistical interpretation of their results, especially when discussing the relationship between ASC items and right anterior insula activity. Please avoid statements like "tended to have". Additionally the lack of significant mediation in the presented figure (Fig 3d) makes it difficult to determine whether the results support a mediation effect, like it is discussed later. Thirdly, the focus on the amygdala and insula cortex is narrow, as ketamine has also been shown to affect other brain regions. From functional connectivity studies effects of ketamine on medial frontal and parietal brain regions are known. Here of interest is that ketamine induced ASC did not show correlations with FC networks despite the task negative network (e.g. doi: 10.1016/j.nicl.2018.05.037). Finally, more explanation and literature citation should be provided for researchers who are not familiar with the mediation analysis method used in the study.

REVIEWER COMMENTS

Reviewer #1 (Remarks to the Author):

1. This is an interesting analysis of 13 healthy volunteers who underwent ketamine infusion and MRI scanning. The authors link dissociative symptoms induced by ketamine as defined by clusters from the CADSS and ASCs with insula and amygdala activity in response to a face processing task. Strengths include the novel research design, strong methods and the use of multiple ketamine doses. Limitations include the small sample size and the generalizability of findings from a healthy volunteer sample to understand the antidepressant response to ketamine. Additionally, as the authors state, the causal mechanisms of dissociation on functional neuroimaging results cannot be determined.

Response: We thank the reviewer for the positive feedback about the strengths of our design, methods, and use of multiple ketamine doses. We highlight that we have now included our prior power analysis, that indicates we have adequate power to detect a meaningful signal in the present small sample, enhanced by the within subjects repeated measures design (Supplementary page 1, lines 12-26).

2. My primary concern is the extension of these results to the antidepressant effects of ketamine. Work by Nugent (2018), Reed (2018, 2019) and Evans (2018) have shown diverging brain responses to ketamine in healthy volunteers, using both resting state and emotional tasks. The relationship of antidepressant response to CADSS experience is also divergent as healthy volunteers often show a transient dysphoric effect of ketamine. Please include additional discussion of the limitations of using healthy volunteer samples to draw conclusions about antidepressant response in depressed patients, using this literature.

Response: The reviewer raises important limitations and we have included as recommended additional discussion of the limitations of using a healthy volunteer sample to draw conclusions about antidepressant response. We thank the reviewer for highlighting the important work of Nugent, Reed and Evans which is an omission in our original submission. We have substantially revised the text to cite this work in the Introduction and Discussion sections. In this reviewed text, we have emphasized the need for future research that expands our current findings in a direct comparison of healthy volunteers and depressed patients. Further, our revised text also refers to the inconsistent findings of antidepressant response to CADSS experience in healthy volunteers and depressed patients (page 18, lines 7-26). We reproduce the revised text below:

“A fifth limitation relates to the extent to which findings from healthy volunteers generalize to an understanding about antidepressant response to ketamine. Previous functional neuroimaging studies observe inconsistent neural effects of ketamine across groups of healthy individuals and patients with MDD. Both groups have been found to show a similar emotional valence-related effect of ketamine versus placebo on ACC subregion activity elicited by anger versus happy in a facial emotion task, along with opposing effects on another ACC subregion and frontal cortex for post-ketamine versus post-placebo (Reed et al 2018¹). Healthy individuals

and patients with MDD have been found to show opposing effects on activity in the temporal gyri and precuneus/posterior cingulate during an implicit emotion task following ketamine versus placebo (Reed et al 2019²). In the resting state, both groups have been observed similar effects of ketamine on gamma power in the ACC and insula (Nugent et al., 2019³) while patients with MDD specifically have been found to show a normalization of insula to default mode connectivity (Evans et al 2018⁴). These previous studies have examined neural changes two days after ketamine infusion and were not designed to assess the acute neural effects immediately following ketamine infusion as in the present study. However, the variability in of findings across healthy and MDD groups highlights the need for future investigations that expand upon the present findings to include clinical participants, enabling a direct comparison of the acute neural effects of ketamine in healthy controls with the acute antidepressant response. Such future investigations would also be important for understanding the relationship of dissociation, assessed by the CADSS and other measures, to the clinical antidepressant response, given that dissociative experiences can also be more intense or extended in MDD (e.g., Nugent et al., 2019³)."

3. My other concern is the "meaning" of the dissociative experience as either causal or an epiphenomenon. Ketamine has an impact across multiple biological systems, so it's difficult to know whether correlated changes in heart rate, ketamine metabolites or gamma power would have had a similar effect on amygdala and brain activity. The factor scores developed from these dissociative scales are so interrelated that it is difficult to parse whether one symptom has a distinct effect from the others. Please include more discussion of why dissociation may be distinctly related to emotional changes after ketamine, as compared to other potential biomarkers.

Response: We thank the reviewer for raising these points, and we will address them one at a time.

First, we entirely agree with the reviewer that ketamine has an impact across multiple biological systems, and dissociative effects of ketamine could be observed in changes in measures in other systems, including heart rate, ketamine metabolites, and gamma power. Given this, it could follow that changes in heart rate, ketamine metabolites, or gamma power might have a similar effect on amygdala and insula brain activity. To address this point, we have substantially expanded the depth of our analyses to include physiological measures. In the design of this study, we did actually monitor heart rate and blood pressures but had not reported them in the current manuscript given we had considered its scope to already be rather wide. However, thanks to the reviewer, now we see the importance of reporting these data to enable more specific interpretation of the results. In the expanded analyses, we tested for heart rate and blood pressure as additional potential physiological mediators of ketamine's effects on amygdala and insula activity. The findings indicate that neither heart rate nor blood pressure are significant mediators. We have added discussions of these non-significant findings in the revised manuscript (page 17, line 26; page 18, lines 1-7) and reproduced it here.

“Fourth, because ketamine has an impact across multiple biological systems, we recognize that there might be additional mediators of the activation of the insula in response to threat, including physiological measures⁵. Although we have shown that physiological measures of heart rate and blood pressure are not significant mediators of change in insula activity in our sample, future studies are needed to test additional potential mediating measures including ketamine metabolites and resting-state magnetoencephalography (MEG) gamma power⁶.”

Descriptions of our method and the non-significant results have been reported in the revised Supplemental Methods (Supplementary page 9 lines 2-6), and in the revised Supplemental Results (Supplementary page 10 lines 23-26). These results have also been reported in the revised Supplemental Table 3 and is reproduced below with the extract of these non-significant results for heart rate and blood pressure.

Extract from revised Supplemental Table 3 (Supplementary page 16)

DV		Mediator	ACME	
			Coef	p-value
Neural activity in response to threat versus neutral	Right anterior insula	Heart rate	-0.02	0.97
		SBP	0.14	0.18
		DBP	0.14	0.18
	Right amygdala	Heart rate	-0.06	0.74
		SBP	0.04	0.5
		DBP	0.01	0.86

DV = Dependent Variable; ACME = Average Causal Mediation Effects; SBP = systolic blood pressure; DBP = diastolic blood pressure; Coef = partially standardized coefficient; †: $p < 0.1$; *: $p < 0.05$; **: $p < 0.01$.

Second, although we did not measure ketamine metabolites or gamma power in our current design, we recognize the importance of these measures as highlighted in our first point above. As outlined in the revised text above, we highlight this is an important area of study in the future.

Third, we also agree with the reviewer that the factor scores developed from dissociative scales are interrelated between subjects (inter-subject) reflecting the way the scales were developed. We realize from the reviewer’s feedback that we did not make it sufficiently clear that in our analysis we were emphasizing intra-subject correlations, rather than inter-subject correlations. We have shown when looking at intra-subject correlations, we can parse different factor scores from the dissociative scales and that the factors mediating the effects of ketamine on insula activity did not show intra-subject correlations. We refer to these factor scores as subcomponents of dissociation. Specifically, depersonalization did not positively correlate with amnesia or anxiety (Repeated Measures Correlation $pFDR > 0.05$). We have clarified these

findings in the Revised Supplemental Methods and Results (Supplementary page 9, lines 8-17; Supplementary page 11, lines 2-10). The revised text is also reproduced below:

“Intra-subject correlation analysis of subcomponents of dissociation and other altered states of consciousness

To explore the direct association between dissociation and other altered states of consciousness, repeated-measures correlation analyses were run using the `rmcorr` package (<https://cran.r-project.org/package=rmcorr>). Unlike conventional inter-subject correlation analysis, the repeated measures correlation analysis calculates intra-subject correlations, which in our case, reflects how different subcomponents of dissociation and other altered states of consciousness co-vary across visits. ... FDR correction was implemented to control for false positives.

Intra-correlation of subcomponents of dissociation and other altered states of consciousness

Although the dissociation subcomponents depersonalization and amnesia were correlated with derealization (Repeated Measures Correlation; depersonalization to derealization $r = 0.93$, $pFDR < 0.001$; amnesia to derealization $r = 0.66$, $pFDR = 0.05$), they were not correlated with each other (Repeated Measures Correlation $r = 0.52$, $pFDR = 0.13$), indicating that they may map onto different factors of ketamine-related experiences. As expected, greater amnesia was associated with greater fear/anxiety, measured by the 5-Dimensional Altered States of Consciousness (5D-ASC) rating scale (Repeated Measures Correlation $r = 0.75$, $pFDR = 0.02$).”

Fourth, we address the reviewer’s point about including more discussion of why dissociation may be distinctly related to emotional changes after ketamine, as compared to other potential biomarkers. We have expanded the discussion to address the reviewer’s comment by drawing on evidence from the PTSD literature (page 12, lines 23-26; page 13, lines 1-5). We reproduce the revised text below:

“Specifically, induction of the depersonalization aspect of dissociation might be an essential ingredient in the mechanisms by which ketamine alleviates negative affective brain states. Ketamine effects on depersonalization, or the sensation of detaching from one’s body, might be accompanied by detachment from negative affective states of emotional pain including depression⁷. This suggestion draws on findings in PTSD in which dissociation is understood to allow for detachment from negative emotional states⁸ and pain perception⁹. Dissociation in PTSD is also associated with enhanced insula and amygdala reactivity during fMRI of the same nonconscious social threat stimuli as used in the present study¹⁰.”

Reviewer #2 (Remarks to the Author):

This manuscript by Hack et al. describes the findings of a study using mediation analyses to

investigate the effects of ketamine-induced altered states of consciousness on brain activation in two regions (bilateral Anterior Insula and Amygdala) involved in the processing of affective material. In a double-blind within-subject design a total of 13 healthy adults received three temporally separate infusions of saline, 0.05mg/kg and 0.5mg/kg ketamine in a random order. Before and after the infusion the CADSS was administered to assess dissociative symptoms followed by an fMRI scan to probe neural responses to implicit affective faces by means of the FEET. Subsequently, the 5D-ASC was employed retrospectively to assess blissful state, anxiety and impaired control and cognition. A linear mixed effects model was used to test the respective hypotheses. Mediation analysis showed that more pronounced dissociative depersonalization attenuates the activity of the right anterior insula in response to threat faces. Furthermore, dissociative amnesia and anxiety (trend) was found to amplify the effect of ketamine on anterior insula activity in response to fearful faces. However, no significant results were shown for the mediating effect of ASCs on amygdala reactivity.

Major points:

1. Please report descriptive statistics and effect sizes for all statistical tests conducted.

Response: We thank the reviewer for these two suggestions, and we address them in order.

Regarding descriptive statistics, we now report mean and standard deviation (SD) of neuroimaging and non-neuroimaging measures in the revised Supplemental Table 1 (Supplementary page 14). The table is also reproduced below.

Suppl. Table 1. Comparison of post-hoc paired t-test results between the placebo, 0.05 mg/kg and 0.5 mg/kg conditions

Measures		Placebo	0.05 mg/kg	0.5 mg/kg	0.05 mg/kg vs. Placebo			0.5 mg/kg vs. Placebo			0.5 mg/kg vs. 0.05 mg/kg		
		Mean (SD)	Mean (SD)	Mean (SD)	T	p	d	T	p	d	T	p	d
ASCs Hypothesized to Relieve Negative Affective Brain States	Depersonalization	0 (0)	0.52 (1.22)	13.22 (16.69)	1.48	0.17	0.45	2.85	0.02*	0.82	2.87	0.02*	0.86
	Derealization	0.67 (1.30)	0.69 (1.36)	14.72 (13.89)	-0.05	0.96	0.02	3.69	0.003**	1.06	3.59	0.004**	1.08
	Bliss	4.59 (13.27)	6.11 (9.41)	25 (28.13)	0.39	0.70	0.12	2.62	0.02*	0.76	2.27	0.04*	0.68
ASCs Hypothesized to Exacerbate Negative Affective Brain States	Amnesia	0 (0)	1.39 (3.24)	9.03 (13.01)	1.48	0.17	0.45	2.50	0.03*	0.72	1.97	0.07	0.60
	Anxiety	2.04 (4.57)	2.57 (4.66)	15.38 (11.00)	1.19	0.26	0.36	5.30	<0.001***	1.53	4.94	<0.001***	1.49
	Impaired control and cognition	2.46 (4.58)	2.89 (5.67)	23.56 (15.05)	0.38	0.71	0.12	5.24	<0.001***	1.51	4.86	<0.001***	1.47
Neural activity in response to threat versus neutral	Right anterior insula	-0.06 (0.68)	-0.28 (0.33)	0.56 (0.32)	-0.49	0.64	0.27	3.22	0.009**	1.02	6.50	<0.001***	2.30
	Right amygdala	-0.29 (0.78)	0.11 (0.48)	0.68 (0.50)	1.56	0.15	0.48	4.05	0.002**	1.28	3.30	0.009**	0.92

Abbreviation: ASCs = Altered States of Consciousness; SD = standard deviation; T = T score from paired t-tests; p = p value from paired t-tests; d = Cohen's d effect size from pairwise contrasts; *: p < 0.05; **: p < 0.01; ***: p < 0.001.

Second, regarding the effect sizes of statistical tests, for linear mixed models examining ketamine’s dose-dependent changes, because we are interested in the standardized mean differences between the drug conditions, we have now calculated the Cohen’s *d* for all pairwise contrasts between drug conditions. This information is included in the revised Supplemental Table 1 (as reproduced above) and is reported in the main text for the 0.5 mg/kg vs. placebo condition of primary interest (page 7, lines 21-23; page 8, lines 1-2; page 9, lines 16-17, 22-25; page 10, lines 6-9). For mediation analysis, partial standardized beta coefficient values were used as a measure of effect size; these are reported in the original main Figures 1d, 2d, and 3d and in the revised Supplemental Table 3 (Supplementary page 16). The revised Supplemental Table 3 is reproduced and referred to in more detail in response to point 2 below.

2. Please also report the statistics of all non-significant findings.

Response: We thank the reviewer for alerting us to our omission; that we had not reported non-significant findings. In the revised manuscript, we have added an additional Supplemental Table 3 (Supplementary page 16) which reports a summary of both the significant and the non-significant results from all mediation analyses. In this table, we also include the standardized beta coefficient values of the ACME mediation model, reflecting the effect size of the mediation.

Suppl. Table 3. Summary of results from mediation analysis

DV		Mediator	ACME		ADE		Total Effect	
			Coef	p-value	Coef	p-value	Coef	p-value
Neural activity in response to threat versus neutral	Right anterior insula	Depersonalization	-0.39	0.004**	1.52	<0.001	1.13	<0.001
		Derealization	-0.22	0.3	1.2	<0.001	0.99	<0.001
		Bliss	0.13	0.41	0.86	<0.001	0.98	<0.001
		Amnesia	0.32	0.04*	0.65	0.01	1.00	<0.001
		Anxiety	0.46	0.08*	0.69	0.07	0.99	<0.001
		Impaired control and cognition	0.28	0.3	0.7	0.04	0.98	<0.001
		Heart rate	-0.02	0.97	0.84	0.01	0.83	<0.001
		SBP	0.14	0.18	0.68	<0.001	0.82	<0.001
		DBP	0.14	0.18	0.68	<0.001	0.82	<0.001
	Right amygdala	Depersonalization	0.04	0.85	1.06	<0.001	1.13	<0.001
		Derealization	0.27	0.12	0.82	<0.001	0.99	<0.001
		Bliss	0.16	0.19	0.93	<0.001	0.98	<0.001
		Amnesia	0.23	0.10	0.86	<0.001	1.00	<0.001
		Anxiety	0.21	0.39	0.89	<0.001	0.99	<0.001
		Impaired control and cognition	0.29	0.25	0.71	<0.001	0.98	<0.001
		Heart rate	-0.06	0.74	0.99	<0.001	0.83	<0.001
		SBP	0.04	0.5	0.88	<0.001	0.82	<0.001
		DBP	0.01	0.86	0.91	<0.001	0.82	<0.001

DV = Dependent Variable; ACME = Average Causal Mediation Effects; ADE = Average Direct Effects; SBP = systolic blood pressure; DBP = diastolic blood pressure; Coef = partially standardized coefficient estimated; +: $p < 0.1$; *: $p < 0.05$; **: $p < 0.01$.

3. In the introduction you stated that detachment, as a component of dissociation, may exert an ameliorative effect on negative affective and pain states. Did you find evidence for that assumption beyond the antidepressant effects of ketamine?

Response: In the Introduction, our intention was to highlight that depersonalization and derealization aspects of dissociation induced by ketamine involve the sensation of detaching from one's body and surroundings, and that this sensation might also be accompanied by detachment from negative affective states of emotional pain including depression. We have now added other lines of evidence in the discussion (page 12, lines 23-26; page 13, lines 1-5) and reproduced the text here.

“Specifically, induction of the depersonalization aspect of dissociation might be an essential ingredient in the mechanisms by which ketamine alleviates negative affective brain states. Ketamine effects on depersonalization, or the sensation of detaching from one's body, might be accompanied by detachment from negative affective states of emotional pain including depression⁷. This suggestion draws on findings in PTSD in which dissociation is understood to allow for detachment from negative emotional states⁸ and pain perception⁹. Dissociation in PTSD is also associated with enhanced insula and amygdala reactivity during fMRI of the same nonconscious social threat stimuli as used in the present study¹⁰.”

4. Please define “positive and negative (brain) states” more clearly.

Response: In the revised text of the Methods, we have now defined positive and negative affective brain states more clearly (page 24, lines 2-6). The revised text is reproduced below:

“We use the term positive affective brain states to mean neural activity that is evoked by positively valenced emotional stimuli, and that may be associated with subjective positive experiences, whereas we use the term negative affective brain states to refer to neural activity that is evoked by negative emotional stimuli such as social threat, and that may be linked with subjective negative experiences such as anxiety.”

5. Please report behavioral data (button presses per condition) from the FEET and add analyses to control that ketamine did not change the subliminal thresholds.

Response: We thank the reviewer for this helpful suggestion to report behavioral data from the Facial Expressions of Emotion Test (FEET) and to consider whether ketamine might change subliminal thresholds.

In response to this suggestion, we report behavioral accuracy and reaction time for button press data for FEET. Linear mixed models were used to test ketamine's dose-dependent effect on each of these two behavioral measures as a dependent variable and with dose (placebo, 0.05 mg/kg or 0.5 mg/kg) as the fixed effect. No significant dose-dependent change was detected for accuracy or reaction time for neutral, threat, or threat vs neutral conditions. We have now included this finding in our main text Results (page 10, lines 9-10) and reproduce it here:

“There were no significant ketamine dose-dependent changes in behavioral performance measures of accuracy (number of correct presses) and reaction time (Suppl. Results).”

This report of the behavioral data findings is also visualized in the revised Supplemental Figure 3, and this figure is reproduced below:

Supplemental Figure 3. Behavioral performances during the nonconscious Facial Expression of Emotion Task (FEET)

Accuracy (number of correct presses) under Threat (a.), Neutral (b.), and Threat vs Neutral (c.). Average reaction time (RT) of the correct presses under Threat (d.), Neutral (e.), and Threat vs Neutral (f.).

Senior author Williams' prior work has demonstrated that reaction times are a measure of the implicit influence of the valence of nonconsciously presented facial emotion stimuli¹¹. In this regard, the reaction time of implicit behavioral performance (button pressing to face stimuli

regardless of emotional valence) is an index of the subliminal perception thresholds. Because ketamine was not found to have a dose-dependent effect on implicit reaction time, including for active dose versus placebo, our conclusion is that ketamine did not impact these thresholds.

As additional sensitivity analyses, we included these behavioral data as covariates of no interest when looking at the dose-dependent effect on ketamine on neural activation measured with fMRI. These covariates did not change the significance of the original results, further indicating that ketamine did not impact the subliminal thresholds for measurement of neural activity. These results are reported in the revised main manuscript (page 10, lines 11-13) and in the Supplemental Methods (Supplementary page 8, lines 12-16), and Supplemental Results (Supplementary page 10, lines 6-10). The text is also reproduced here.

“Additional sensitivity analysis showed consistently the same dose-dependent effect of ketamine on anterior insula or amygdala activity when controlling for the effect of behavioral performance (**Suppl. Methods and Results**).

Sensitivity analysis of dose-dependent effects of ketamine on anterior insula and amygdala activity in response to threat confounded by behavioral performance

As additional sensitivity analyses, we added the accuracy (button press number) and mean RT as covariates of no interest when examining ketamine’s dose-dependent effect on FEET activations in selected ROIs.

Ketamine’s effect on insula and amygdala activity remains the same when controlling for behavior performance

In additional sensitivity analyses controlling for button presses and mean RT, we similarly observed dose-dependent effects of ketamine on the activity of right anterior insula ($F_{2,22} = 9.14, p = 0.001$) and right amygdala ($F_{2,22} = 6.68, p = 0.005$) evoked by threat faces.”

Minor points:

1. The inclusion criteria require participants to have used ketamine at least two times previously to the administration in the study. What are the reasons for defining this criterium?

Response: This criterion was required by our local IRB. We have now clarified this under “Methods: Participants” (page 19, line 16).

2. Please move all quantitative ASC to the supplement.

Response: Given the quantitative dose-dependent effect on ASC is one of our major findings, we believe the reviewer might be suggesting to put all qualitative ASC quotes into the supplement. We appreciate the opportunity to clarify that we would hope to keep the quotes in the main text, as they bring the terminology of ketamine-induced dissociation to life and would provide our general readers, especially those from other fields, an intuitive

understanding of what these scales were measuring. In the current manuscript we do have summarized all quotes into Supplemental Table 2 (Supplementary page 15).

3. Did you control for potential order effects of the 3 conditions in your within-subjects design?

Response: We thank the reviewer for raising this important question about control of order effects. In our design, the primary control of the potential order effect was through the random assignment of drug visit orders. However, given the sample size, it remains possible that randomization did not fully control for all sources of potential order effect. Thus, we undertook an explicit sensitivity test of drug visit order across subjects using linear mixed models. We did not identify any significant effects of order. This additional analysis is reported in the revised Supplemental Methods (Supplementary page 8 lines 18-26) and revised Supplemental Results (Supplementary page 10 lines 12-21), as reproduced below.

“Sensitivity analysis of dose-dependent effects of ketamine on dissociation and altered states of consciousness, and on anterior insula and amygdala activity in response to threat after controlling for the order of drug visits

To test whether the order of drug visits had an impact on our current finding, we created a categorical variable representing the six drug visit orders: i) placebo, 0.05 mg/kg, 0.5 mg/kg; ii) placebo, 0.5 mg/kg, 0.05 mg/kg; iii) 0.05 mg/kg, placebo, 0.5 mg/kg; iv) 0.05 mg/kg, 0.5 mg/kg, placebo; v) 0.5 mg/kg, placebo, 0.05 mg/kg; and vi) 0.5 mg/kg, 0.05 mg/kg, placebo. As another set of sensitivity analyses, we added the categorical order variable to all linear mixed models used for testing ketamine induced dose-dependent effects.

Ketamine’s effect on altered states of consciousness and dissociation, and on anterior insula and amygdala activity in response to threat remains the same when controlling for the order of drug visits

We still observed a significant dose-dependent effect of ketamine in dissociative depersonalization ($F_{2,76} = 11.10, p < 0.001$), dissociative derealization ($F_{2,76} = 19.27, p < 0.001$), dissociative amnesia ($F_{2,76} = 7.57, p = 0.001$), altered states of bliss ($F_{2,25} = 8.66, p = 0.001$), anxiety ($F_{2,25} = 36.57, p < 0.001$) and impaired control and cognition ($F_{2,25} = 31.75, p < 0.001$). Similarly, we observed dose-dependent effects of ketamine on increasing right anterior insula ($F_{2,36} = 9.23, p < 0.001$) and right amygdala activity ($F_{2,19} = 18.46, p < 0.001$) evoked by threat faces.”

4. What kind of coefficients are depicted in Fig. 1.d., Fig. 2.d. and Fig. 3.d.?

Response: The coefficients depicted in Fig. 1d, Fig. 2d, and Fig. 3d are beta coefficient values estimated from mediation analyses with standardized M and Y variables. These coefficient values shown in the line of text “Mediation of X on Y via M: ...” in Fig. 1d, Fig. 2d, and

Supplementary Fig. 4d indicate the indirect effect of X on Y via M. They reflect the effect size of the mediation model, represented as ketamine induced change in neural activity in the magnitude of the standard deviation transmitted by the change in ketamine induced dissociation or altered state of consciousness. In the revised text of the main manuscript, we have included this specification (page 25, lines 14-26, page 26, line 1), and this revised text is reproduced below. Additionally, we have expanded our explanations of the mediation model which are referred to in more detail in response to Reviewer 3 Minor point 4 below.

“In our design using ACME, the intervention is designated as the independent variable ‘X’ (0.5 mg/kg versus placebo), dissociation or other altered state of consciousness is the mediator variable ‘M’ and emotion task-elicited neural activity is the dependent variable ‘Y’. ... The X variable was coded as 0 (placebo) and 1 (0.5 mg/kg), and the M and Y variables were standardized. This approach enabled us to estimate effect size for the average direct effects of X on Y, indirect effects of X on Y transmitted via M and the total effect for the overall ACME model¹². The primary effect size of interest is the indirect mediation effect, represented as ketamine induced change in neural activity in the magnitude of the standard deviation transmitted by the change in ketamine induced dissociation or altered state of consciousness. Because these effect sizes were standardized they can be compared across studies and interpreted similarly to correlations¹³, < 0.2 is considered a weak effect; ≥ 0.2 and ≤ 0.5 , a moderate effect; and > 0.5, a strong effect.”

5. Please indicate more clearly in Fig. 1,2 and 3 that findings only apply for the RIGHT anterior insula and for threat faces only

Response: We appreciate this suggestion and have added text in Fig. 1d, 2d, and 3d (now Supplementary Fig. 4) to clearly indicate the findings only apply for the RIGHT side of anterior insula and for threat faces only.

6. Please improve the understanding of the following sentences: “Because the n-back emotion task relies on cognitive regulation/suppression of limbic reactivity, it is also possible that less participant-level dissociation, relevant to the esketamine formulation, is required for attenuation of this reactivity.” (p. 13) and “Indeed, successful antidepressant treatment lowers insula reactivity to negative stimuli³⁸, and our prior work using the same FEET task of current study in a novel behavior intervention study has found that early change in insula reactivity to negative stimuli predicts future treatment outcomes as a function of treatment ³⁹, and another prior pharmacotherapy study also using the same FEET has shown that reactivity of another critical affective brain region – the amygdala – to negative emotional stimuli is predictive of antidepressant response⁴⁰.” (p.14).

Response: We have substantially revised this section. To improve understanding of the findings using the n-back task, we now refer specifically to the purpose of the design to assess ketamine’s neural effects on emotion stimuli in the context of subject’s performing a verbal

working memory task. We have also incorporated citations to the work of Reed et al., 2018 and Reed et al., 2019. The revised text (page 13, lines 14-26; page 14, lines 1-2) is reproduced below:

“The emotion task deployed in our trial is designed to evoke reactivity of these regions, including under drug conditions. Prior trials imaging facial emotion-evoked activity after two days post-ketamine have observed increases in amygdala for individuals with MDD, but decreases in the insula, and opposing patterns of change for healthy individuals^{1,2}. In studies deploying imaging immediately after an IV infusion, healthy individuals also show reductions in acute ketamine-induced insula and amygdala reactivity, compared to a non-IV baseline, while performing a task that assesses the effects of emotional stimuli on a cognitive process, verbal working memory^{14,15}. There are several possible explanations for these variations in findings. The direction of effect might differ with both clinical status of participants and the duration of the post-infusion period and we address this further under potential study limitations. In the present study, the use of a randomized placebo comparison controlled for the likely arousing and contextual effects of the IV line, which by itself may increase limbic reactivity. Our randomized placebo control also addressed the likelihood that findings may reflect the natural attenuation of neural reactivity when using a non-randomized, non-IV baseline, as was used in these prior designs^{14,15}.”

Reviewer #3 (Remarks to the Author):

The article discusses a study on the effects of ketamine on the brain's affective circuitry. The study found that ketamine induced both positive and negative altered states of consciousness (ASCs) and that the specific type of ASC induced by ketamine may be crucial to its antidepressant effects. The study's findings suggest that previous mixed results on the relationship between dissociation and antidepressant response may be due to differences in the subcomponents of dissociation experienced.

Overall, the study provides important insights into the complex relationship between ketamine-induced ASCs and neural changes in the brain's affective circuitry. The findings suggest that the effects of ketamine on negative affective states are dependent on the specific aspects of ASCs induced by the drug. This information could be useful in developing more targeted and effective treatments for depression and other mood disorders. While the study provides valuable insights into the effects of ketamine on altered states of consciousness and its potential therapeutic use, there are some concerns that need to be addressed.

Response: We thank the Reviewer for the positive feedback.

1. Firstly, the sample size of nonclinical participants used in the study is relatively small, which may limit the generalizability of the results to clinical populations. A power analysis could be provided to support the adequacy of the sample size.

Response: We thank the review for the suggestion of reporting our a priori power analysis to justify the current sample size (N = 13). Our power analysis was based on the effect sizes reported in previous literature and factors in our repeated measures design. This analysis demonstrates we have adequate power to detect significant effects on our primary measures of interest and we have included this information in the revised Supplementary Methods (Supplementary page 1, lines 12-26).

“To estimate the sample size for detecting ketamine’s effect on our primary neural dependent measures of negative affect circuit activation in response to emotional stimuli, we drew on prior reported effect sizes for a prior study in which imaging was similarly undertaken immediately following ketamine infusion in a repeated measures design¹⁵. In this prior study the effect size reported for left and right amygdala activity was $\eta^2 = 0.43$ and $\eta^2 = 0.31$ respectively. For primary self-reported dissociation effects of interest, we similarly drew on prior reported effect sizes in a separate study¹⁶, and the effect size was $\eta^2 = 0.38$. Using the most conservative effect size of $\eta^2 = 0.31$, which was converted to the Cohen’s *f* of 0.67, we conducted a power analysis using G*Power Version 3.1.9.6¹⁷ for sample size estimation. The sample size needed to detect an effect on the amygdala or insula with $\alpha = .05$ and at least 95% power for a within-subject design with two repeated imaging measurements is $n=10$. Here we overestimated the needed sample size by including only two dose conditions (the placebo and the ketamine at 0.5 mg/kg) to match previous study designs. Still, the obtained sample size of $n = 13$ is adequate to detect ketamine’s dose-dependent effects on primary measures of interest.”

2. Secondly, the authors should be more precise in the statistical interpretation of their results, especially when discussing the relationship between ASC items and right anterior insula activity. Please avoid statements like “tended to have”. Additionally the lack of significant mediation in the presented figure (Fig 3d) makes it difficult to determine whether the results support a mediation effect, like it is discussed later.

Response: We thank the reviewer for raising this point regarding being more precise in the statistical interpretation of our results and for highlighting specific examples. Regarding the “tended to have” statement, we have removed it from the revised manuscript. To be more precise we have stated, when examining the associations between ASC items and insula activity, specifically the direct comparison of high and low groups stratified based on depersonalization or amnesia, the effect sizes and the statistical significance in the Main Text (page 11, lines 1-7, 14-20).

Because the lack of significant mediation of anxiety originally presented in Fig 3d is not primary to the results, we have removed it from the revised manuscript and instead reported this non-significant mediation in the Supplemental Results (Supplemental Fig. 4, Supplementary page 23).

3. Thirdly, the focus on the amygdala and insula cortex is narrow, as ketamine has also been shown to affect other brain regions. From functional connectivity studies effects of ketamine on

medial frontal and parietal brain regions are known. Here of interest is that ketamine induced ASC did not show correlations with FC networks despite the task negative network (e.g. doi: 10.1016/j.nicl.2018.05.037).

Response: We agree with the reviewer that ketamine has also been shown to affect other brain regions in addition to the amygdala and insula. Our rationale for focusing on the amygdala and insula was drawn on two inter-related lines of prior research. The first relates to our specific focus on ketamine's effects on task-elicited neural activity, rather than resting FC. Arguably, ketamine's effects on regions of activity elicited under specific task conditions is more circumscribed than on task-free intrinsic network connectivity. We draw on the senior author's prior findings with the same nonconscious task (FEET) demonstrating that this task engages the negative affect circuit with major nodes of activity in the amygdala and insula¹⁸, and that these specific regions of activity are modulated by intervention in disorders such as depression¹⁹. Second, we draw on a prior finding using acute ketamine administration demonstrating specific effects on the amygdala elicited by an emotion task, albeit without comparison to a placebo control¹⁴.

Nonetheless, we have also included in the revised analyses, the major prefrontal nodes that define the negative affect circuit in addition to the amygdala and insula; specifically the subgenual anterior cingulate cortex (sgACC) and dorsal anterior cingulate cortex (dACC). Inclusion of these ACC regions further builds on prior established methods in randomized controlled designs¹⁹, and allows us to interpret our findings in the context of an additional study of ketamine's effects on emotion task-elicited neural activity, which report effects on the ACC (albeit within 2-days of ketamine infusion and not immediately afterward; Reed et al 2018¹). These revised analyses did not reveal a significant dose-dependent effect of ketamine for sgACC or for dACC activity (page 10, lines 13-16; reproduced below). Thus, we did not include these regions in mediation analyses and retain the focus of our interpretation on amygdala and insula activity.

"We did not observe any effects of ketamine ... when examining sgACC and dACC as evoked by threat or happy faces."

4. Finally, more explanation and literature citation should be provided for researchers who are not familiar with the mediation analysis method used in the study.

Response: We thank the reviewer for the suggestion of improving our manuscript by including more explanation and literature citation for the mediation analysis. We have now added citations and the explanation of how mediation works, including the parameters generated from mediation analysis. These revisions in the text of the main manuscript (page 25, lines 1-26; page 26, lines 1-2) are reproduced below:

“Mediation analysis using an averaged causal mediation effect model

To address our third objective—to test whether specific aspects of ketamine-induced dissociation and other ASCs mediate the effect of dose on acute changes in neural activity during emotional processing—we utilized the Averaged Causal Mediation Effect (ACME) mediation model^{20,21}. ACME models were used to test our working hypotheses that: 1) ketamine will reduce neural activity reflecting relief of negative affective states, mediated by depersonalization and derealization from the CADSS and blissful state from the 5D-ASC; and 2) ketamine will increase neural activity reflecting exacerbation of negative affective states, mediated by dissociative amnesia from the CADSS, as well as anxiety and impaired control and cognition from the 5D-ASC. ACME is a method for dissecting the total effect of an intervention (in this case ketamine) into direct effects on neural activity and indirect effects on neural activity that are transmitted via a mediator (in this case dissociation or other altered state of consciousness). It addresses the limitation of analysis designs that incorrectly treat intermediate or mediator variables as confounding factors. In our design using ACME, the intervention is designated as the independent variable ‘X’ (0.5 mg/kg versus placebo), dissociation or other altered state of consciousness is the mediator variable ‘M’ and emotion task-elicited neural activity is the dependent variable ‘Y’. We implemented the ACME using the mediation package (<https://cran.r-project.org/web/packages/mediation/index.html>) in R, in combination with the lmer package to account for the repeated design. The X variable was coded as 0 (placebo) and 1 (0.5 mg/kg), and the M and Y variables were standardized. This approach enabled us to estimate effect size for the average direct effects of X on Y, indirect effects of X on Y transmitted via M and the total effect for the overall ACME model¹². The primary effect size of interest is the indirect mediation effect, represented as ketamine induced change in neural activity in the magnitude of the standard deviation transmitted by the change in ketamine induced dissociation or altered state of consciousness. Because these effect sizes were standardized they can be compared across studies and interpreted similarly to correlations¹³, < 0.2 is considered a weak effect; ≥ 0.2 and ≤ 0.5, a moderate effect; and > 0.5, a strong effect. Statistical significance (p value) was generated using bootstrapping procedures with 1000 resamples.

REFERENCES

1. Reed, J.L., *et al.* Ketamine normalizes brain activity during emotionally valenced attentional processing in depression. *Neuroimage Clin* **20**, 92-101 (2018).
2. Reed, J.L., *et al.* Effects of Ketamine on Brain Activity During Emotional Processing: Differential Findings in Depressed Versus Healthy Control Participants. *Biol Psychiatry Cogn Neurosci Neuroimaging* **4**, 610-618 (2019).
3. Nugent, A.C., *et al.* Ketamine has distinct electrophysiological and behavioral effects in depressed and healthy subjects. *Mol Psychiatry* **24**, 1040-1052 (2019).
4. Evans, J.W., *et al.* Default Mode Connectivity in Major Depressive Disorder Measured Up to 10 Days After Ketamine Administration. *Biol Psychiatry* **84**, 582-590 (2018).

5. McIntyre, R.S., *et al.* Synthesizing the Evidence for Ketamine and Esketamine in Treatment-Resistant Depression: An International Expert Opinion on the Available Evidence and Implementation. *Am J Psychiatry* **178**, 383-399 (2021).
6. Farmer, C.A., *et al.* Correction: Ketamine metabolites, clinical response, and gamma power in a randomized, placebo-controlled, crossover trial for treatment-resistant major depression. *Neuropsychopharmacol* **45**, 1405 (2020).
7. Alexander, L., Jelen, L.A., Mehta, M.A. & Young, A.H. The anterior cingulate cortex as a key locus of ketamine's antidepressant action. *Neuroscience & Biobehavioral Reviews* **127**, 531-554 (2021).
8. Oathes, D.J. & Ray, W.J. Dissociative tendencies and facilitated emotional processing. *Emotion* **8**, 653-661 (2008).
9. Duckworth, M.P., Iezzi, T., Archibald, Y., Haertlein, P. & Klinck, A. Dissociation and posttraumatic stress symptoms in patients with chronic pain. *International Journal of Rehabilitation and Health* **5**, 129-139 (2000).
10. Felmingham, K., *et al.* Dissociative responses to conscious and non-conscious fear impact underlying brain function in post-traumatic stress disorder. *Psychol Med* **38**, 1771-1780 (2008).
11. Williams, L.M., *et al.* Explicit identification and implicit recognition of facial emotions: I. Age effects in males and females across 10 decades. *J Clin Exp Neuropsychol* **31**, 257-277 (2009).
12. Igartua, J.J. & Hayes, A.F. Mediation, Moderation, and Conditional Process Analysis: Concepts, Computations, and Some Common Confusions. *Span J Psychol* **24**, e49 (2021).
13. Acock, A.C. *A gentle introduction to Stata*, (Stata press, 2008).
14. Scheidegger, M., *et al.* Effects of ketamine on cognition-emotion interaction in the brain. *Neuroimage* **124**, 8-15 (2016).
15. Scheidegger, M., *et al.* Ketamine administration reduces amygdalo-hippocampal reactivity to emotional stimulation. *Hum Brain Mapp* **37**, 1941-1952 (2016).
16. Nugent, A.C., *et al.* Ketamine has distinct electrophysiological and behavioral effects in depressed and healthy subjects. *Molecular Psychiatry* **24**, 1040-1052 (2019).
17. Faul, F., Erdfelder, E., Lang, A.-G. & Buchner, A. G*Power 3: A flexible statistical power analysis program for the social, behavioral, and biomedical sciences. *Behavior Research Methods* **39**, 175-191 (2007).
18. Goldstein-Piekarski, A.N., *et al.* Mapping Neural Circuit Biotypes to Symptoms and Behavioral Dimensions of Depression and Anxiety. *Biol Psychiatry* **91**, 561-571 (2022).
19. Goldstein-Piekarski, A.N., *et al.* Early changes in neural circuit function engaged by negative emotion and modified by behavioural intervention are associated with depression and problem-solving outcomes: A report from the ENGAGE randomized controlled trial. *Ebiomedicine* **67**, 103387 (2021).
20. Imai, K., Keele, L. & Tingley, D. A general approach to causal mediation analysis. *Psychol Methods* **15**, 309-334 (2010).
21. Imai, K., Keele, L., Tingley, D. & Yamamoto, T. Unpacking the Black Box of Causality: Learning about Causal Mechanisms from Experimental and Observational Studies. *American Political Science Review* **105**, 765-789 (2011).

REVIEWERS' COMMENTS

Reviewer #1 (Remarks to the Author):

Appreciate the thoughtful and comprehensive response from the authors. My concerns about the HV sample have been addressed in the limitations section.

Reviewer #2 (Remarks to the Author):

The edits made to the manuscript are acceptable.

Reviewer #3 (Remarks to the Author):

All comments of my review are sufficiently addressed. I want to congratulate the authors for the thorough revisions and I would recommend publication of the revised manuscript.

REVIEWER COMMENTS

Reviewer #1 (Remarks to the Author):

Appreciate the thoughtful and comprehensive response from the authors. My concerns about the HV sample have been addressed in the limitations section.

Response: We are glad that the Reviewer found our response to be thoughtful and comprehensive and that the Reviewer's concerns about the HV sample have been addressed.

Reviewer #2 (Remarks to the Author):

The edits made to the manuscript are acceptable.

Response: We thank the Reviewer for reviewing our revised manuscript.

Reviewer #3 (Remarks to the Author):

All comments of my review are sufficiently addressed. I want to congratulate the authors for the thorough revisions and I would recommend publication of the revised manuscript.

Response: We are delighted that all the Reviewer's comments have been addressed and thank the Reviewer for their kind words.